

# Errors in stereoscopic retrievals of cloud top height for single-layer clouds.

Jesse Loveridge[1,†], Larry Di Girolamo[1]

[1]Department of Atmospheric Sciences, University of Illinois, Urbana, 61801, USA
[†]Now at Department of Atmospheric Sciences, Colorado State University, CO, USA.

*Correspondence to*: Jesse Loveridge (Jesse.Loveridge@colostate.edu)

**Abstract.**

Multi-angle stereoscopic methods are a promising means for retrieving high-resolution cloud volumes and their temporal evolution. Stereoscopic retrievals assume that light emerges from localized points on a surface. We assess the errors
introduced by this assumption using synthetic measurements at various wavelengths, solar-viewing geometries, and spatial resolutions generated by applying a 3D radiative transfer model to an ensemble of 841 (8 km)$^2$ cloud fields of varying fractional cover, cloud-top bumpiness, microphysics, and optical depth. We show that stereoscopic retrievals of cloud top height (CTH) have biases that vary from -175 m to +20 m as the cloud-edge extinction profile becomes sharper and absorption increases, all when mean visible cloud optical depth is greater than 5, and with little dependence on instrument
resolution between 50 m and 250 m. Stereo CTH fields are smoother than the truth when CTH variability is concentrated at small spatial scales, viewing angles are oblique, and absorption is weak. We attribute this effect to both the smoothing effect of multiple scattering which is stronger at wavelengths with weak absorption, and the ill-posed nature of the retrieval in the presence of non-uniform CTH over the stereo matching window. The standard deviation of stereo CTH errors increases from 25 m to 200 m as the standard deviation of CTH increases to 200 m over the (8 km)$^2$ domain. More than 50% of stereo
retrievals from two different 50 m resolution stereo viewing pairs of (0°, +38°) and (-38°, 0°) are consistent to within 30 m over (500 m)$^2$ regions for clouds with standard deviation of CTH less than 200 m. We analysed airborne lidar observations and found that 75% of shallow cumulus and all stratocumulus have standard deviations of CTH less than 200 m over 8 km transects. These results support the application of time-differenced stereoscopic cloud top height retrievals for the remote sensing of high-resolution cloud dynamics as well as macrophysics.

## 1 Introduction

Observations of cloud macrophysical properties such as their coverage, base height, top height and vertical extent provide insight into the role of cloud spatial organization in modulating the cloud radiative effect (Giuseppe and Tompkins, 2003; O'Hirok and Gautier, 1998), the efficiency of surface precipitation formation (Radtke et al., 2023; Smalley and Rapp, 2021), and the vertical transport of momentum, moisture, and energy (Garrett et al., 2018; Neggers et al., 2019; Peters et al., 2020,



2021). The coupling between these three processes of turbulence, precipitation and radiation is central to the uncertainty in cloud radiative feedback in a changing climate (Kazil et al., 2024; Vial et al., 2017; Zhao et al., 2016). Accurate measurements of cloud boundaries provide indirect constraints on both the longwave cloud radiative feedback and the shortwave cloud radiative feedback (Aerenson et al., 2022; Klein et al., 2017; McKim et al., 2024; Vial et al., 2023). Precise measurements of cloud boundaries are also the first step to direct measurements of how cloud volumes change with time

(Dandini et al., 2022b), which provides an indirect constraint on processes such as entrainment. Measurements of cloud boundaries are valuable for forecasting the availability of solar energy (Peng et al., 2015) due to the strong relationship between 3D cloud geometry and shortwave radiation (Cahalan et al., 2005; Davies, 1978; Killen and Ellingson, 1994; Kobayashi, 1991; O'Hirok and Gautier, 1998). This strong relationship also means that measuring cloud boundaries (i.e., 3D cloud geometry) constitutes the first step for accurate remote sensing of cloud optical and microphysical properties from

shortwave radiance measurements (Chambers et al., 1997; Ewald et al., 2019; Fielding et al., 2014; Iwabuchi and Hayasaka, 2003; Levis et al., 2020; Loveridge et al., 2023b). The broad utility of cloud boundary measurements highlights the importance of continuing to develop, validate, and improve our methods of measuring cloud boundaries.

The boundaries of clouds are regularly retrieved by spaceborne, airborne and ground-based remote sensing instruments

including lidar (Vaughan et al., 2009), radar (Mace and Zhang, 2014), passive spectro-radiometry in the thermal infrared (Baum et al., 2012) and near-infrared (Yin et al., 2020), and stereoscopic techniques using multi-angle passive imagery typically collected in the solar part of the spectrum (Beekmans et al., 2016; Muller et al., 2007, 2002). Stereoscopic techniques are one of the most promising remote sensing modalities for retrieving cloud boundaries from space. This is due to their ability to achieve high (< 50 m) resolution and precision retrievals of cloud boundaries from multiple viewing angles

(Castro et al., 2020; Dandini et al., 2022b), enabling reconstruction of the photo hull of a cloud volume over wide swath widths (e.g. ~100s of km) (Dandini et al., 2022b; Muller et al., 2007, 2002). This capability strongly complements the ability of spaceborne lidar and radar to precisely determine the boundaries of optically thin layers and the interior structure of cloud volumes, though only over 2D swathes, at relatively coarse resolution (~1 km) (Mace and Zhang, 2014) and with much higher instrumental expense. The precision of stereoscopic techniques and their sensitivity to clouds that are undetectable by

Doppler cloud radar (Battaglia et al., 2020; Burns et al., 2016; Lamer et al., 2020), makes them a promising basis for retrieving the evolution of cloud volume boundaries associated with updrafts as weak as ~1 m/s (Dandini et al., 2022b).

The long history of applying stereoscopic techniques to spaceborne instruments such as the Multiangle Imaging SpectroRadiometer (MISR) (Muller et al., 2002) and the Along Track Scanning Radiometer (ATSR) (Muller et al., 2007)

has led to a good understanding of the strengths and weaknesses of the technique. Stereoscopic cloud top height retrievals are invariant to radiometric calibration, making them powerful for trend detection (Davies et al., 2017). They are also highly precise when sufficient texture exists to detect correspondences between images using block matchers (Marchand et al., 2007; Mitra et al., 2021; Muller et al., 2002; Naud et al., 2004, 2005). A dominant contribution to the uncertainty in single-



platform retrievals of cloud top height using stereoscopic techniques is the difficulty of separating the parallax signal of
cloud top height from the temporal evolution of the clouds (Horváth and Davies, 2001; Mitra et al., 2021; Mueller et al.,
2017). This uncertainty contribution has also been recognized from airborne, single-platform stereo at higher spatial
resolution (Kölling et al., 2019; Volkmer et al., 2024a, b). The issue of temporal evolution can be mitigated through the use
of multiple measurement platforms, which can also enable the detection of change over short time scales (Dandini et al.,
2022b). When clouds are sufficiently optically thin or homogeneous to otherwise not produce a strong texture, then the
stereo height can fail to retrieve the cloud boundary or retrieve a position that is interior to the actual cloud boundary
(Marchand et al., 2007; Mitra et al., 2021; Naud et al., 2004, 2005). This is especially apparent in multi-layered cloud
conditions where optically thin cirrus overly optically thicker, and highly texture lower cloud layers. However this effect is
even apparent in textured, optically thick single-layered clouds, where it leads to a negative bias in height on the order of -
100 m with respect to lidar measurements (Mitra et al., 2021). This negative bias has been termed the stereo-opacity bias
(Mitra et al., 2021).

The stereo opacity bias is important because it is a cloud-dependent systematic error and is one of the few components of the
uncertainty budget of stereo retrievals that has yet to be fully characterized. For sample-rich datasets such as satellite remote
sensing, it is the systematic errors which are the most important for controlling the measurement uncertainty in trend-
detection or other scientific analysis. The stereo opacity bias is unique in the uncertainty budget of a stereoscopic retrieval as
all other components such as geo-registration errors, matching error, and wind-sensitivity (if applicable), are all traceable
and well-characterized for established retrievals such as MISR's stereo cloud top height retrieval (Mitra et al., 2021; Mueller
et al., 2017). The stereo-opacity bias may be quite important in some applications despite its apparently small magnitude in
optically thick, boundary layer clouds. For example, the magnitude of the bias can be similar to the diameter of a shallow
cumulus cloud (De Vera et al., 2024; Zhao and Di Girolamo, 2007; Zhao and Austin, 2005); a stereoscopic volume
reconstruction with a ~100 m bias along the line of sight of each view would lead to a drastic underestimation of cloud
volume in such cases.

Recent estimates of the accuracy of stereoscopic cloud top height retrievals diverge in the magnitude of the stereo-opacity
bias with values of -15 m to -126 m for optically thick boundary layer clouds (Dandini et al., 2022a; Kölling et al., 2019;
Mitra et al., 2021; Volkmer et al., 2024b). Part of this range arises from differences in the definition of the cloud boundary in
model-based studies using Large Eddy Simulations (Dandini et al., 2022a; Volkmer et al., 2024b), which is a somewhat
ambiguous concept at the microphysical level (Di Girolamo and Davies, 1997; Koren et al., 2008). Given these diverging
results, and the limited case studies that have been examined, it is unclear how the magnitude of the stereo opacity bias
varies by cloud type.





Basic radiative transfer theory indicates that the stereo opacity bias will depend on cloud type. Stereoscopic retrievals are derived from a simple approximation – that all the light that has emerged from a surface does so in a highly localized fashion near the surface. Indeed, this is a requirement in the original definition of the bi-directional reflectance distribution function (BRDF) that is meant to define the intrinsic scattering properties of surfaces (Nicodemus et al., 1977). The presence of significant sub-surface scattering or emission, which is common to many man-made and natural objects, leads to the failure of this simple approximation. Volumetric radiative transfer in heterogeneous media can create features in the observed multi-angle imagery that do not lie on the boundary of the media. When a stereoscopic method is applied to find the best-fitting surface at which image features can be co-registered, that surface will be interior to the cloud, producing the stereo-opacity bias. The degree to which this occurs will depend on the 3D volumetric structure of the cloud, hence cloud type.

The objective of this study is to systematically evaluate the simple modelling approximation that underlies stereoscopic retrievals of cloud boundaries and leads to the stereo opacity bias. We do this by quantifying how the errors in stereoscopic height retrievals change with instrument resolution, solar-viewing geometry and wavelength, for an ensemble of synthetic clouds with different cloud optical thicknesses, microphysics, and degrees of heterogeneity. This approach is designed to complement existing modelling studies, which have focused on the capabilities of particular observing systems and utilized only conservatively scattering visible radiation (Dandini et al., 2022b; Volkmer et al., 2024a). To help understand the significance of the performance of the simulated retrievals with cloud type, we also utilize airborne lidar observations from the Cloud, Aerosol and Monsoon Processes Experiment (CAMP²Ex) field campaign (Reid et al., 2023) to better understand how representative our simulations are for the cumuliform clouds sampled during the campaign. We discuss the implications of our results for the design of future remote sensing systems and the interpretation of existing records of stereoscopic cloud top height retrievals. We comment on the feasibility of retrieving temporal changes in cloud top height (related to vertical velocity) using the Tandem Stereo Cameras concept for NASA's Atmosphere Observing System (Braun et al., 2022). We also provide recommendations for developing stereo algorithms tailored for cloud remote sensing applications.

## 2 Methodology

### 2.1 Synthetic cloud fields

We use stochastically generated cloud fields as described in Loveridge & Di Girolamo (2024). The stochastic cloud generator outputs 3D liquid water content and effective radius at 50 m resolution over (8 km x 8 km x 2 km) domains. We choose a resolution of 50 m to resolve the 3D radiative transfer for the typical range of volume extinction coefficient of liquid clouds (20 m to 200 m) (Kokhanovsky, 2004). There are three key features of our stochastic cloud generator that make it appropriate for our study. First, we can systematically sample the properties of the cloud (e.g., optical thickness or degree of cloud-top bumpiness). Of particular importance, is our ability to prescribe the scale-dependence of the cloud-top height variability i.e., the bumpiness, which is extremely difficult to do with a dynamical model such as a Large Eddy Simulation



(LES). Second, we can construct the cloud to be simple so that there is a unique cloud-top height for each $(x, y)$ position to facilitate a simpler interpretation of the results. Lastly, the method is computationally trivial in comparison to LES so we can examine large ensembles of clouds with ease.

The stochastic cloud generator takes four quantities as input. These include the cloud-mean visible optical depth, $\bar{\tau}$, the cloud-mean droplet number concentration, $N_0$, the standard deviation of the geometric thickness ($f_H$) and the slope of the power law describing the power spectrum of horizontal variance, $\beta$. Other properties of the cloud field are constrained by various assumptions. The cloud is assumed to have a constant cloud-base-height of 450 m and a mean geometric thickness of 350 m. A truncated-normal distribution of cloud geometric thickness is assumed. Cloud geometric thickness is distributed horizontally by filtering 2D white noise to have a power spectrum following a prescribed power law (with slope $\beta$). The cloud fraction is calculated based on the standard deviation of the geometric thickness according to the truncated-normal distribution assumption (Considine et al., 1997). A quasi-adiabatic assumption is used to derive the vertical variability of geometric-optics volume extinction coefficient and effective radius from the cloud-mean droplet number concentration and the optical depth and geometric thickness of each column (Grosvenor et al., 2018). A gamma droplet size distribution with an effective variance of 0.07 is assumed. The droplet number concentration at each cloudy grid point that is adjacent to a clear grid point is replaced by its value after smoothing with a gaussian kernel with a width of 30 m. The liquid water content and geometric-optics volume extinction coefficient are then recalculated assuming that the droplet effective radius remains unchanged, mimicking heterogeneous mixing near cloud edge (Beals et al., 2015; Lehmann et al., 2009). This component of the stochastic cloud generator is important as it means that the droplet number concentration parameter $N_0$ also controls the magnitude of the cloud-edge gradient in extinction coefficient and is likely to affect the stereo opacity bias. This feature is varied independently of the magnitude of the extinction coefficient, which is controlled by $\bar{\tau}$ and $f_H$.

We generate 841 cloud fields according to a Latin Hypercube sampling (Stein, 1987) of the four input parameters, $\bar{\tau}$, $N_0$, $f_H$, $\beta$. The cloud-mean optical depth is sampled logarithmically from 5 to 20. The cloud-mean droplet number concentration is sampled logarithmically from 60 cm$^{-3}$ to 300 cm$^{-3}$. The standard deviation of the geometric thickness is sampled linearly from (24 m to 260 m) with a corresponding variation of cloud fraction between 50% and overcast. The power-law slope of the horizontal variance is sampled linearly from -3 to -2. The resulting cloud fields have maximum column optical depths of ~220 and maximum cloud-top droplet effective radius of ~25 $\mu m$. Loveridge and Di Girolamo (2024) showed that these stochastically generated cloud fields spanned the range of cloud geometry observed during the Cloud System Evolution in the Trades (CSET) field campaign (Albrecht et al., 2019) and other sources (Boers et al., 1988; Loeb et al., 1998) for ~8 km long transects with at least 50% cloud cover. It was also shown that the internal variability of the cloud fields is consistent with in situ measurements from the Rain in Cumulus over Ocean (RICO) and VOCALS-Rex field campaigns (Boutle et al., 2014). These facts give us confidence that we have a reasonable representation of cloud structure for evaluating stereoscopic





retrievals of cloud top height for boundary layer clouds, while also providing a diverse set of simulations to evaluate the cloud-dependence of the retrieval errors.

## 2.2 3D radiative transfer simulations

We perform 3D radiative transfer simulations using the Spherical Harmonics Discrete Ordinates Method (SHDOM) (Evans, 1998) with periodic horizontal boundary conditions. We simulate the band-averaged radiance at several MODIS spectral bands using the REPTRAN gas absorption parameterization (Gasteiger et al., 2014) to explore a range of cloud-remote sensing wavelengths for stereo retrievals. We simulate MODIS (Barnes et al., 2003) Band 2 (0.86 $\mu m$), Band 6 (1.6 $\mu m$), Band 7 (2.1 $\mu m$), Band 20 (3.7 $\mu m$) and Band 31 (11 $\mu m$). Hereafter, we refer to the bands by their nominal wavelength. We perform our 3D radiative transfer simulations for every cloud at three different solar zenith angles (30°, 45°, 65°). We use the tropical, midlatitude summer, and midlatitude winter standard atmospheres (Anderson, 1986) to calculate molecular and aerosol scattering and absorption at the respective solar zenith angles. The covariation of atmosphere and solar zenith angle is done to achieve a more realistic partitioning of thermal and solar radiation at 3.7 $\mu m$ for sampling by a satellite in a sun-synchronous orbit. We include absorption by water vapor, ozone, $CO_2$, $N_2O$, CO, $CH_4$, $O_2$, $N_2$, and Rayleigh scattering by all gases. Aerosol scattering and absorption are included based on the OPAC maritime tropical aerosol type (Hess et al., 1998).

The simulations use an ocean surface BRDF, using the 6S model (Vermote et al., 1997) for all bands except 11 µm. The 6S model uses a wind speed of 8 ms[-1] and a pigment concentration of 0.08 mg m[-3]. At 11 $\mu m$ we assume the surface BRDF is Lambertian with an emissivity of 0.979 (Niclòs et al., 2005). The surface temperature is that of the lowest level in the atmosphere. The vertical base grid of the SHDOM simulations is that of the cloud (50 m) up to 2 km and then is that of the AFGL standard atmosphere above that (1 km resolution up to 25 km and 2.5 km resolution above that). The SHDOM solver uses 16 zenith angle discrete ordinate bins and 32 azimuthal discrete ordinate bins at 0.86 $\mu m$, 1.6 $\mu m$, 2.1 $\mu m$ and the solar portion at 3.7 $\mu m$, while just 8 and 16 at 11 $\mu m$ and 12 and 24 for the thermal portion at 3.7 $\mu m$. Within SHDOM, the solution accuracy is set to $5 \times 10^{-5}$ and the grid splitting accuracy is set to 0.03 for solar simulations (for unit solar flux), and to 0.03 times the surface's blackbody radiance in thermal simulations.

We calculate radiances using SHDOM at 50 m sampling across the domain top. We assume a linear interpolation of the radiance field and use this to calculate synthetic radiances over pixels of 50 m, 100 m, and 250 m resolution. We calculate these radiances at thirteen viewing zenith angles in a single azimuthal plane, $\pm70.5°$, $\pm60°$, $\pm45.6°$, $\pm38.1°$, $\pm26.1°$ $\pm10°$ and 0°. These angles are the nine of MISR, an additional $\pm38.1°$ pair based on the tandem stereo camera concept proposed for NASA's Atmosphere Observing System (Braun et al., 2022), and a $\pm10°$ pair for sensitivity testing. The



relative azimuth angle used for each simulation is randomly selected from one of four values for each solar zenith angle and cloud.

## 2.3 Stereo Matcher

A stereo matcher is a tool to identify correspondences between images (Bleyer and Breiteneder, 2013). The output of the matcher is a disparity map, which states the number of pixels separating the reference pixel in the reference image to the corresponding feature in the matched image. Image disparities, which have fractional accuracy (e.g., 1.42 pixels), can be converted to a position vector $(x, y, z)$ using the known viewing geometries of the two cameras. The $z$-component of the retrieved position is typically considered to be a point on the outer-boundary of the target (e.g., a cloud top height at x, y). The retrieval of this disparity map is an ill-posed problem. For example, two views won't necessarily see the same features (due to occlusion, for example) and no valid matches may be found. The fraction of successful retrievals, or coverage, of a stereo matcher is therefore as important to consider as the precision of the successful matches.

We utilize the More Global Matcher (MGM) to compute stereo correspondences (Facciolo et al., 2015) between images as in Dandini et al. (2022). The MGM is part of the NASA Ames Stereo Pipeline software for processing satellite imagery (Beyer et al., 2018), making it an established point of reference for assessing stereo-matching quality. The MGM is a regularized block matcher that uses a semi-global optimization method to identify the best-fitting disparity at each point in the reference image by minimizing a cost function. We utilize the census cost function in our matching, which is widely utilized for cloudy images due to its robustness to changes in the magnitude of the brightness, which is important given the strong anisotropy of light scattered by clouds (Fisher et al., 2016). We use the 'v-fit' method to compute sub-pixel disparities. We filter retrievals to require that matching image 1 with image 2 is consistent with matching image 2 with image 1 to within 1 pixel. This filtering removes outliers but reduces successful retrieval coverage.

An important feature of a block/area matcher like MGM is that, while it can skilfully match textures, it relies on an assumption of constant disparity across the matching window to derive the match. This is distinct from a purely global matcher (Tao et al., 2001). This is not a problem when the surface is artificially flat (e.g., a brick wall). However, for other objects like a tree's canopy this is more difficult due to the production of texture through changes in disparity (Goldbergs et al., 2019). Then, for objects that have significant sub-surface scattering like clouds, there can be additional difficulties as the non-local volumetric scattering can cause a decorrelation between the observed texture and the disparity field, as discussed in Section 1.

The MGM has two regularization parameters, which are important for controlling retrieval accuracy. The first regularization parameter, $p_1$, increases the matching cost function that MGM uses to select the best-fitting disparity when disparities between adjacent pixels are larger than 1, to select for solutions that don't have large jumps in disparity. This punishes the



retrieval of non-flat surfaces and is extremely helpful for the retrieval of artificial structures such as buildings. The second regularization parameter, $p_2$, increases the matching cost function when disparities between adjacent pixels are larger than 2. This introduces a smoothing regularization to the retrieval.

In our work, we keep $p_1 = 0$ as we found that non-zero values produced lower precision retrievals for bumpy clouds in our initial sensitivity tests (not shown). We examine sensitivity of the retrieval to $p_2$ for several different values (0, 4, 8, 10, 20) and to the window-size of the census matcher (3 pixels, 5 pixels, 7 pixels). We perform matches between each oblique viewing zenith angle and the nadir view.

## 2.4 Evaluation metrics

To evaluate the stereoscopic retrieval, we first define the ground-truth cloud top height based on the cloud field. SHDOM employs a trilinear interpolation between grid points and so the point at which there is a complete absence of cloud is the first non-cloudy grid point above the cloud. This first vertical location with an absence of cloud is defined as the cloud top height. The ground truth cloud-top-height that is used as reference for comparison to the stereo retrievals is horizontally averaged to the same resolution (e.g., 50 m, 250 m), while ignoring clear sky. This choice for evaluation is not unique. Cloud

top height has scale-dependent variability, such that any finite resolution measurement will always have errors with respect to the unresolved portion of the cloud. For some applications, the mean may be the most relevant statistic while for others the maximum cloud-top-height within the field of view may be more appropriate. For example, the mean has a closer relationship with the effective emission temperature of a cloud as temperature will tend to vary linearly with height over a typical range of cloud-top-height variability that might be un-resolved at 250 m resolution. We highlight that, in cases where

the difference in the definition of cloud-top-height are important for the scientific interpretation of the retrieval, the solution is to increase the resolution of the retrieval so that the phenomenon of interest is resolved.

Errors statistics for remote sensing retrieval algorithms include both sampling error and retrieval error and both need to be characterized for a complete understanding of the performance of a retrieval (Povey and Grainger, 2015). Sampling errors

occur when remote sensing retrievals are only applied on a subset of the sampled data, or are only successful on a subset (Cho et al., 2015). For example, MODIS cloud-top-height retrievals are only reported for locations that the cloud mask has designated as being cloudy (Baum et al., 2012). The differences between the cloud top heights of the pixels on which retrievals are successful and those for which retrievals fail or are not attempted constitutes sampling errors. Meanwhile, differences between the retrieved cloud top height and the ground-truth cloud top height are retrieval error. Categorical

separations of populations of pixels into clear and cloudy are imperfect and will always have some misclassifications (Di Girolamo and Davies, 1997; Frey et al., 2008; Wielicki and Parker, 1992; Yang and Di Girolamo, 2008). The exact definition of cloud vs. clear or the details of the process that leads to retrieval failure or misclassification (Cho et al., 2015; Marchant et al., 2016, 2020) will determine the partitioning of errors between sampling error and retrieval error.



When we want to understand uncertainties in a remote sensing retrieval while the instrument is in operation, we will not have access to a ground-truth designation of cloud vs. clear for each pixel on which to condition an uncertainty model for the retrieval. Instead, we will only have access to an imperfect definition of cloud vs. clear from our measurements. There are four categories in which to assess retrieval error; true positives (true cloud & retrieved cloud), false negatives (true cloud & retrieved clear), false positives (true clear & retrieved cloud), and true negatives (true clear and retrieved clear). Frequently,

studies that assess retrieval errors using simulations only report error results for true positives (Mason et al., 2024) or true cloud (Zhang et al., 2012) which gives an incomplete picture of the error budget. We therefore assess both sampling error and retrieval error by emulating the cloud masking process.

We choose to separate cloudy and clear pixels using a radiometric cloud mask. We use a radiometric cloud mask rather than

the stereo retrieval itself to define cloud and clear categories because the stereo retrieval frequently reports cloud-like heights over clear sky within a few pixels of cloud edge. Figure 1 provides an example of this, where the stereo retrieval (Fig. 1c) has height retrievals in regions that are clear (Fig. 1a) and are correctly flagged as clear by the radiometric cloud mask (Fig. 1d). The radiometric cloud mask is derived from the nadir view at $0.86\ \mu m$ based on a bidirectional reflectance factor threshold of 0.0531 at solar zenith angles of 45° and 65° and a threshold of 0.0594 at a solar zenith angle of 30° due to the

vicinity of the nadir view to the sunglint region. These thresholds correspond to a possibly cloudy designation following the MOD35 cloud mask (Frey et al., 2008). We apply these thresholds at all pixel resolutions. In general, at a solar zenith angle of 30° and resolution of 50 m, the radiometric cloud mask has a true-positive rate of 0.94, a false positive rate of 0.0, a true negative rate of 0.04 and a false negative rate of 0.02. The performance is very similar across resolutions and solar zenith angles. The false negative rate is cloud-dependent, maximizing at 17% of the pixels that do contain some cloud at the

minimum cloud fraction of 50%. Those pixels that are identified as entirely clear by the radiometric cloud mask do tend to have very low ground-truth cloud fractions. To summarize, the radiometric cloud mask is cloud-conservative, it never produces false positives (Yang and Di Girolamo, 2008). The cloud mask preferentially excludes the thinnest clouds which, due to the construction of our cloud fields, are also those with the lowest cloud-top heights (Section 2.1).

For stereo retrievals we also need to consider the sampling bias resulting from failed stereo retrievals in pixels which are designated as cloud by the radiometric cloud mask. Therefore, to evaluate the performance of the stereo retrieval we assess the retrieval accuracy on the pixels masked as cloud with successful cloud top height retrievals and the sampling bias as the difference between the ground-truth mean cloud top height and the mean of the successful retrievals. In other words, the sampling bias includes both bias due to retrieval failure and due to cloud-mask misclassification. As an example, the

sampling bias for the cloud field in Fig. 1 is quantified by subtracting the mean of the non-zero cloud top heights in Fig. 1b from the mean of the valid (non-gray) non-zero cloud top heights in Fig. 1d. We evaluate the fraction of successful retrievals as the fraction of radiometrically-identified cloud with successful retrievals, which we refer to as retrieval coverage. In Fig.



1d, this is the number of non-zero, valid (non-gray) cloud top heights divided by the total number of non-zero cloud top heights.


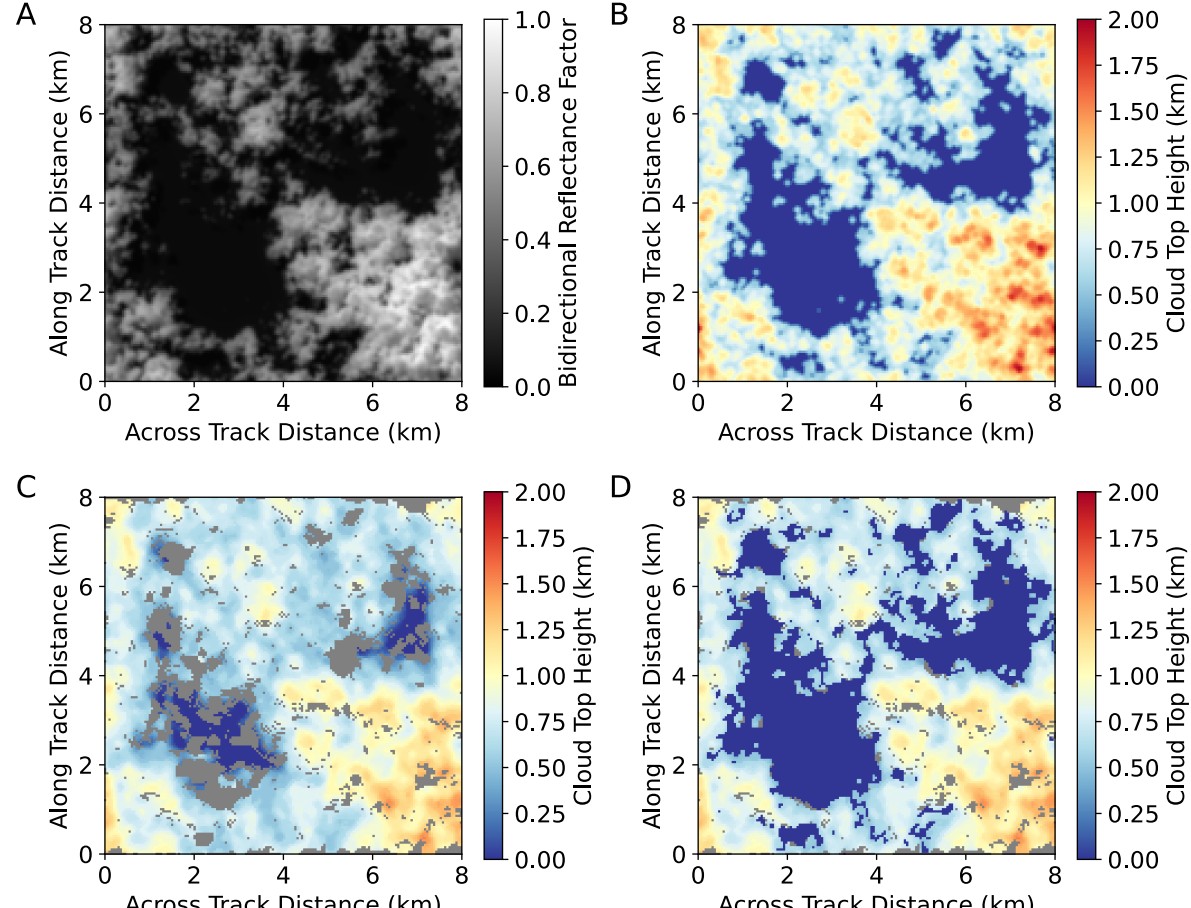

**Figure 1. a) The 50 m nadir bidirectional reflectance factor at 858 nm at a solar zenith angle of 30° for a cloud field with $\bar{\tau} = 14.6$, $\beta = -2.78$, $N_0 = 150$ cm⁻³, and $f_H = 243$ m. b) The ground-truth cloud top height field at 50 m resolution. C) The stereo retrieval from a (26.1°, 0°) pair with failed retrievals shown in gray. D) The stereo retrieval after filtering by the radiometric cloud**
**mask. Pixels identified by the cloud mask as clear are assigned a cloud top height of 0 km.**

We quantify the retrieval accuracy of the successful retrievals for each cloud field using the bias and standard deviation of the errors. Errors in the cloud-top height field are not random. To compute systematic errors in the retrieved cloud top height (other than the bias), we compute the slope parameter of the least-squares regression of the error in the stereo retrieval of 305 cloud top height against the ground-truth cloud top height, which we refer to as the error slope. The error slope is computed



as follows. Given $N$ successful retrievals per cloud field, with cloud top heights $H_{r,i}$, and true cloud top heights $H_{t,i}$, for the $i^{\text{th}}$ retrieval, the error slope parameter, $m$, is computed as

$$m = \frac{N \sum_i H_{t,i}(H_{r,i} - H_{t,i}) - \sum_i H_{t,i} \sum_i H_{r,i}}{N \sum_i H_{t,i}^2 - \left(\sum_i H_{t,i}\right)^2} \qquad \text{(Eq. 1)}$$

Negative error slopes indicate that the retrieved cloud top height field has weaker cloud-top height variability than the ground truth cloud field while positive slopes indicate that the retrieved cloud top height field has more variability than the ground truth.

We also compute statistics of the differences in cloud top height retrievals between different stereo pairs, e.g., the difference between the height retrieved from $(+38.1°, 0°)$ and the height retrieved from $(-38.1°, 0°)$. The error statistics of these pairs
are relevant for determining the ability to detect change, as certain errors (e.g., biases) may be common to both pairs. For these differences, we compute the coverage of shared retrievals as well as their bias, the standard deviation of errors, and median absolute errors.

## 2.5 Data

The focus of our study is to use synthetic cloud fields to assess how stereo retrieval errors vary with cloud type. The relative
frequency of occurrence of the different cloud types in nature is critically important to understand the implications of these results for error statistics in real cloud populations. We use observations from the CAMP$^2$Ex field campaign (Reid et al., 2023) to understand how relevant our simulations are to quantifying uncertainty in stereo retrievals for the climatically important shallow cumulus and congestus clouds sampled during the campaign. We utilize data from the airborne HSRL-2 lidar deployed during the CAMP$^2$Ex field campaign (Reid et al., 2023), as processed in Fu et al., 2022, to quantify the
variability in cloud top height over similar domain sizes as our synthetic cloud fields. The HSRL-2 was deployed on the P-3 aircraft with a maximum altitude of around 8 km and a typical flight speed of 130 ms$^{-1}$ to 200 ms$^{-1}$. The data collection is at 2 Hz, providing an effective resolution of 60 m to 100 m. We computed the standard deviation of cloud top height over a 50 second interval for rough comparison with the $(8\ \text{km})^2$ domain of the synthetic cloud fields. We only use those 50 second intervals which we filtered to contain single-layered, boundary layer cloud by asserting that the maximum cloud top height
should be less than 3 km. All 19 research flights are used.



## 3 Results

### 3.1 Dependence of stereo performance on hyper-parameters of the stereo matcher

We evaluated the median root-mean-square error (rmse) and mean retrieval coverage for the stereo cloud top height retrievals for each combination of the window size and the regularization parameter $p_2$ (Fig. 2). We found that the fraction of

successful stereo retrievals increases with non-zero regularization ($p_2 > 0$), and that rmse decreases for smaller window sizes and larger values of the regularization parameter ($p_2$) (Fig. 2). This trend in performance is common to all clouds, wavelengths, camera pairs, and instrument resolutions. As we describe and discuss in more detail in the following sections, this variation in performance is due to the sensitivity of the stereo precision to variability in cloud top height within the matching window. In the remainder of Section 3, we report in detail the results only for the best-performing configuration,

which uses a window size of 3 pixels and   $p_2 = 20$, to explore how retrieval performance varies with cloud-type, wavelength, camera-pair and resolution. The attached data contains the results for all stereo configurations (Loveridge, 2024). Many of the features of the retrieval performance have common physical explanations, which we provide in Section 4.

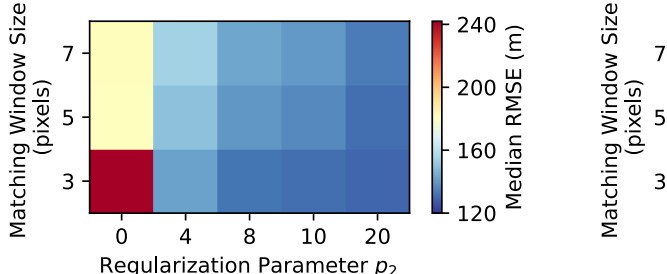
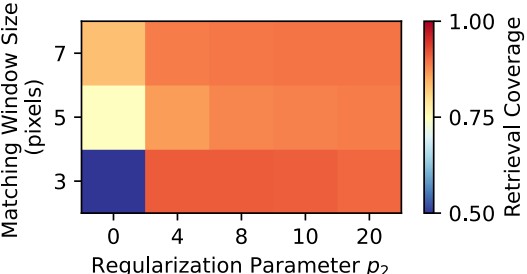

**Figure 2. Left: The variation of the median RMSE across all combinations of wavelength, resolution, and solar-viewing geometry for different matching window size and regularization strength. Right: As above but for the fraction of cloud that has successful retrievals.**

### 3.2 Performance by solar-viewing-geometry, wavelength and resolution

We compute the mean of the error metrics (bias, standard deviation, coverage, error slope) over all clouds and examine their

systematic variations across wavelength, camera pair, resolution, and solar zenith angle. The first quantity we examine is the bias (Fig. 3). The bias is on the order of -100 m and varies only weakly with camera pair, resolution, and wavelength for reflected solar radiation. By contrast, the 11 $\mu m$ channel is the only configuration that has a positive bias, which is reached at viewing zenith angles of 70°. There are notable asymmetries in the bias between camera pairs observing in the forward and backward hemispheres at a solar zenith angle of 30°, especially in the 3.75 $\mu m$ channel. These results highlight the relevance

of scattering in setting the stereo opacity bias, which we discuss in detail in Section 4 in combination with other results. At more oblique solar zenith angles, there is a stronger viewing zenith angle dependence to the bias that is common to all





wavelengths, with biases for the most oblique views that are less than half as large as the biases for the views nearest nadir. To fully attribute this change in bias, we must also consider the change in retrieval coverage with viewing zenith angle and any change in sampling biases, as the set of successful retrievals are not common across viewing angle.

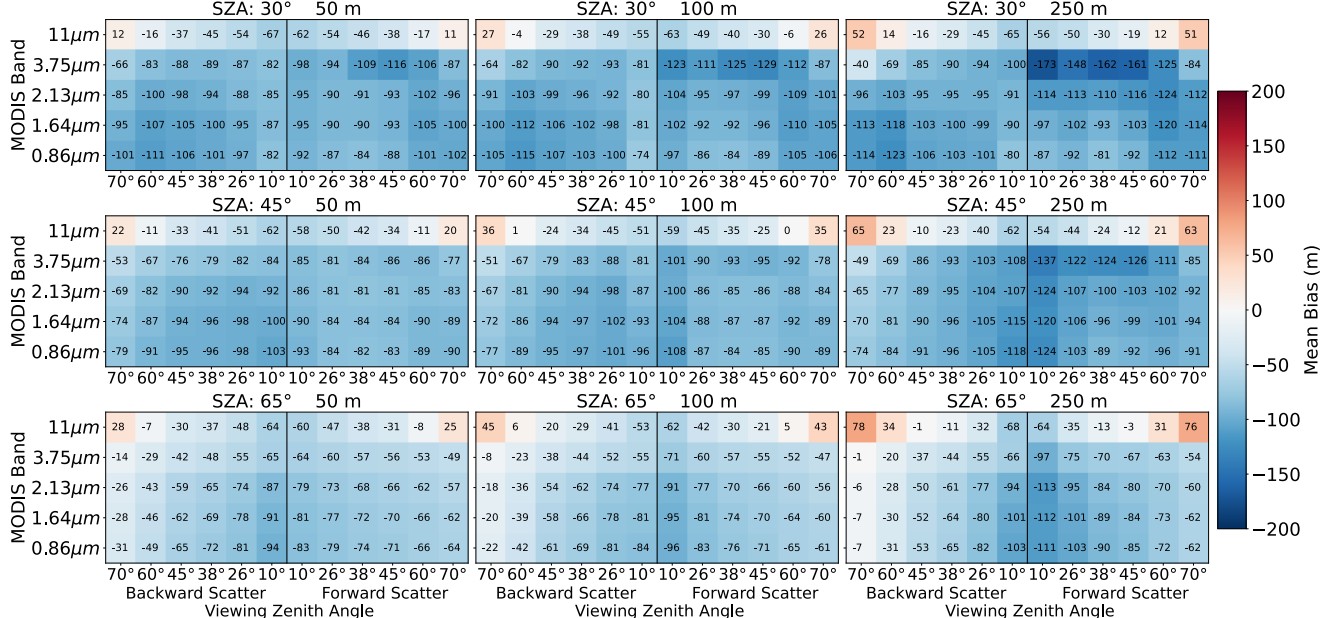

**Figure 3. Each panel shows the variation of the mean bias in the successful stereo retrievals across all cloud fields for different combinations of wavelength and viewing geometry. The different panels show different solar zenith angles (rows) and resolutions (columns).**

Despite the reduction in bias for more oblique views, the large stereo angle between the oblique cameras and the nadir view means that they suffer in their coverage (Fig. 4). The explanation for this behaviour that is common to all instrumental configurations is that it becomes increasingly rare for a feature to be observed by both the nadir and oblique view without obscuration unless the cloud is quite flat. The wavelengths with the most anisotropic scattering are the worst performing in terms of retrieval coverage, a feature that we provide a physical explanation for in Section 4. We find that sampling biases are small (Fig. 5), with their variation by viewing zenith angle being less than half of the viewing zenith angle dependence of the opacity bias. This demonstrates that the change in bias with viewing zenith angle is not due to a preferential failure of retrievals in pixels that would otherwise have more negative biases. We can therefore ascribe this feature a physical cause. In particular, the longer optical paths per unit altitude at more oblique views ensure that image features that are common to both the oblique view and the nadir view emerge from near cloud top.



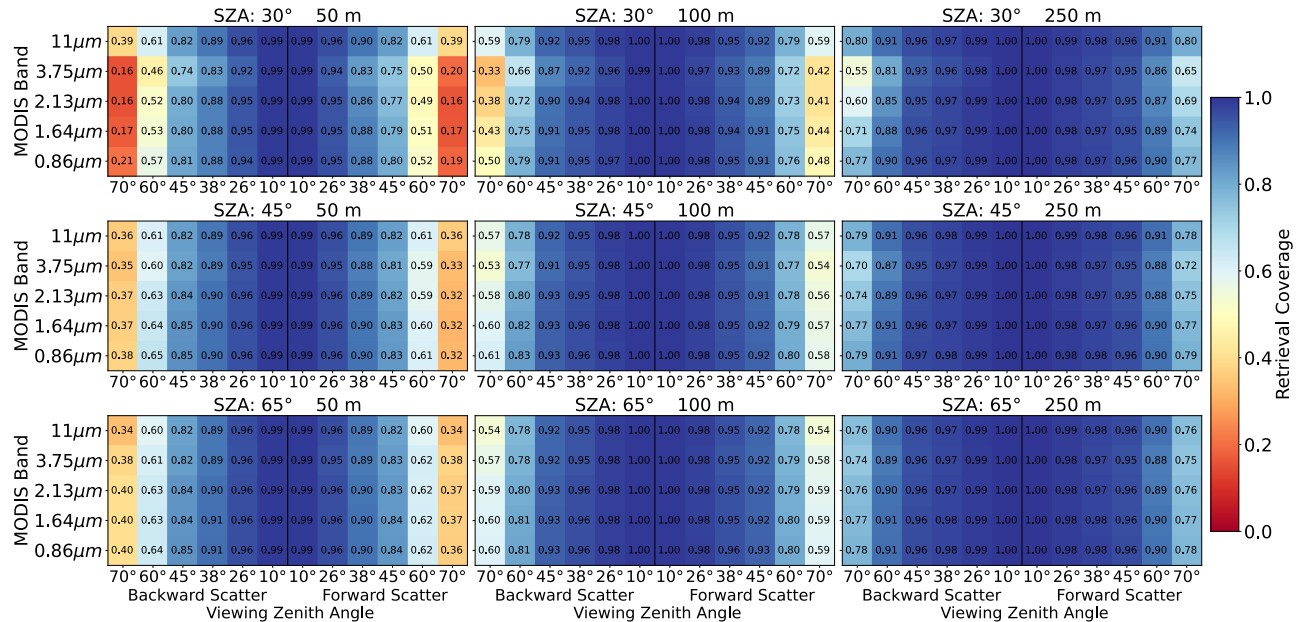

**Figure 4. As in Fig. 3 but for the fraction of cloudy pixels with successful stereo retrievals averaged over all cloud fields. Cloudy pixels are identified using the radiometric cloud mask.**

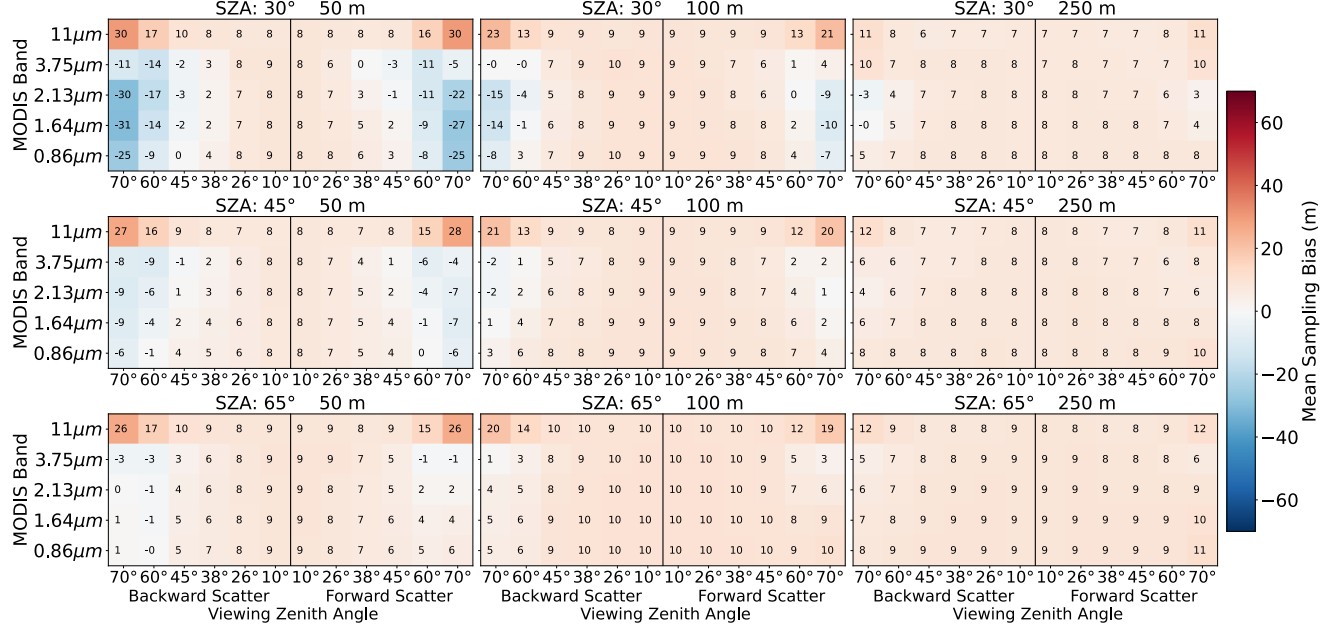

**Figure 5. As in Figure 3 but for the mean sampling bias across all cloud fields.**

The precision of the stereo matches stays stable with changes in viewing zenith angles, particularly at oblique views (Fig. 6). This behaviour is in stark contrast to the bias. The most notable feature in Fig. 6 is that the precision degrades for the 10°



viewing zenith angle pair, especially at coarse resolution. This is because the stereo angle is small enough that the image
disparities are comparable to or much smaller than 1 pixel. This means that the details of the cloud structure are unresolved

and the precision of the retrieval relies entirely on the model for sub-pixel matching (Section 2.3). For a target with constant
ground-truth disparity across the matching window, we would expect the precision to increase with resolution in proportion
to $1/(\tan(\theta_1) - \tan(\theta_2))$, where $\theta_1$ and $\theta_2$ are the viewing zenith angles of a stereo pair in the epipolar plane. We would
therefore expect stereo retrievals to become more precise as viewing angles are more separated, and as resolution increases.
The fact the precision varies only weakly with increasing resolution and viewing zenith angle suggests that variability in the

cloud-top height at a similar scale to the matching window (3 pixels) is degrading the precision of the retrieval. This effect
slows the increase of the precision of stereo retrievals with instrument resolution. This behaviour shows that great care must
be taken when extrapolating matching accuracy to higher resolution instruments.

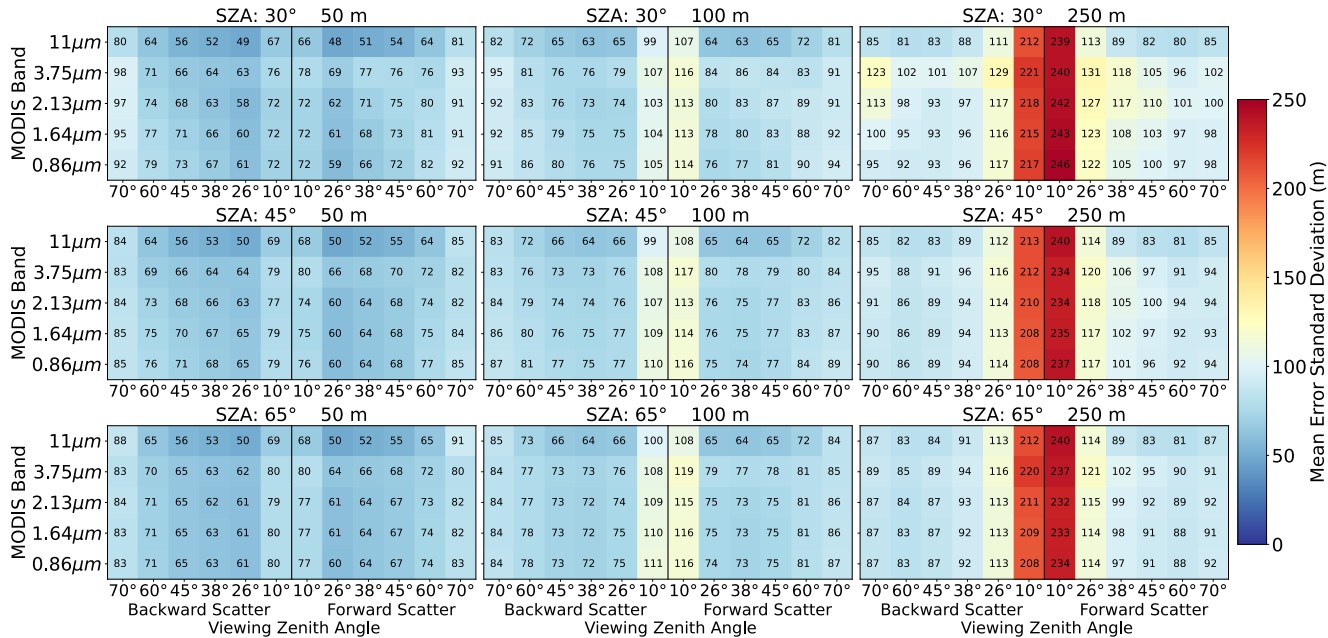

**Figure 6. As in Figure 3 but for the mean error standard deviation across all cloud fields.**

All retrieval configurations retrieve cloud-top-height fields that are smoother than the ground truth, on average. We quantify
this with the error slope; the slope of the stereo height error with respect to the ground-truth cloud top height (Fig. 7).
Negative values of the error slope mean that the retrieved cloud top height field is smoother than the truth. The smoothing
error highlights again the difficulty in retrieving the cloud-top-height variability at a scale commensurate with the matching

window (3 pixels). The error slope is the error metric in which we find the strongest, unambiguous signal of spectral
variability in optical properties. The smoothing error is systematically worse for shorter wavelengths with weaker absorption
and for more oblique views at 50 m and 100 m resolutions. At 250 m resolution, this spectral variation is not as clearly





monotonic, especially at a solar zenith angle of 30°. However, even in this case, the 11 $\mu m$ retrieval still provides the best performing configurations. The strong spectral dependence of this error feature highlights the role of the physics of radiative

transfer, in setting the retrieval error, not just cloud structure and viewing geometry. We explain the physics behind this result in detail in Section 4.

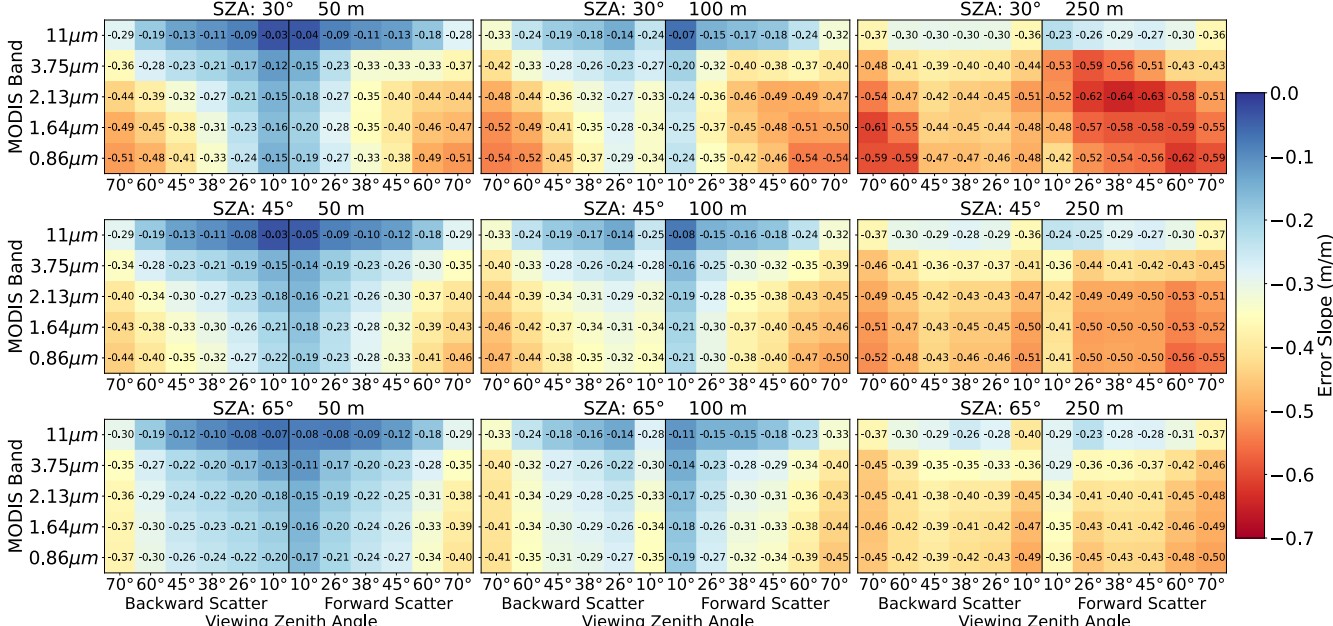

**Figure 7. As in Fig. 3 but showing the mean of the error slope (see text in Section 2.4 for details) across all cloud fields.**

### 3.3 Stereo performance by cloud type

To understand the cause of the errors such as the stereo-opacity bias (Fig. 3) or the smoothing error (Fig. 7) in more detail we examine how these errors vary by cloud type. All analysis presented in this section uses the backscattering 26.1° view angle at 50 m resolution at 0.86 $\mu m$. Results for other wavelengths and viewing pairs are qualitatively similar and can be extrapolated based on Figs. 3 – 7. The standard deviation of the stereo error tends to increase as clouds have more cloud-top height variability and as the proportion of cloud-top height variability at smaller scales is increased (more positive $\beta$) (Fig.

8a). This pattern is quantitatively consistent across wavelength and solar-viewing geometry (e.g., Fig. 6). The change in error standard deviation with cloud type is much larger than any systematic change with wavelength or solar-viewing geometry.

The smoothing error, quantified by the error slope, also strongly varies with $\beta$ and to a lesser extent with the degree of cloud top height variability (Fig. 8b). This demonstrates that when cloud-top height variability is larger at smaller scales, the stereo

retrieval tends to overestimate lower cloud top heights and underestimate the heights of high cloud tops. The variation of the error slope with cloud type shown in Fig. 8b is shifted towards zero for wavelengths with larger absorption and larger solar zenith angle (e.g., Fig. 7).




The strong dependence of height errors with cloud geometry is echoed in the fraction of successful retrievals or retrieval
coverage. The retrieval coverage reduces from around 95% of cloud to as low as 40% of cloud when the standard deviation
of cloud top height reaches 250 m (Fig. 8c). This is largely driven by the increase in obscuration issues and is, like the error
standard deviation, only weakly sensitive to wavelength and solar geometry. Associated with this is a sampling bias, which
is small, being strictly less than 5% of the mean cloud top height in any cloud field (Fig. 8d). The sampling bias tends to be
negative, as retrieval failures tend to be focused on the highest cloud tops. This is compensated by the exclusion of low-
altitude optically thin cloud by the cloud mask. This leads to a dependence of the sampling bias on mean cloud optical depth
and the standard deviation of cloud top height, as these are the parameters of the stochastic cloud generator that control the
occurrence of optically thin cloud.

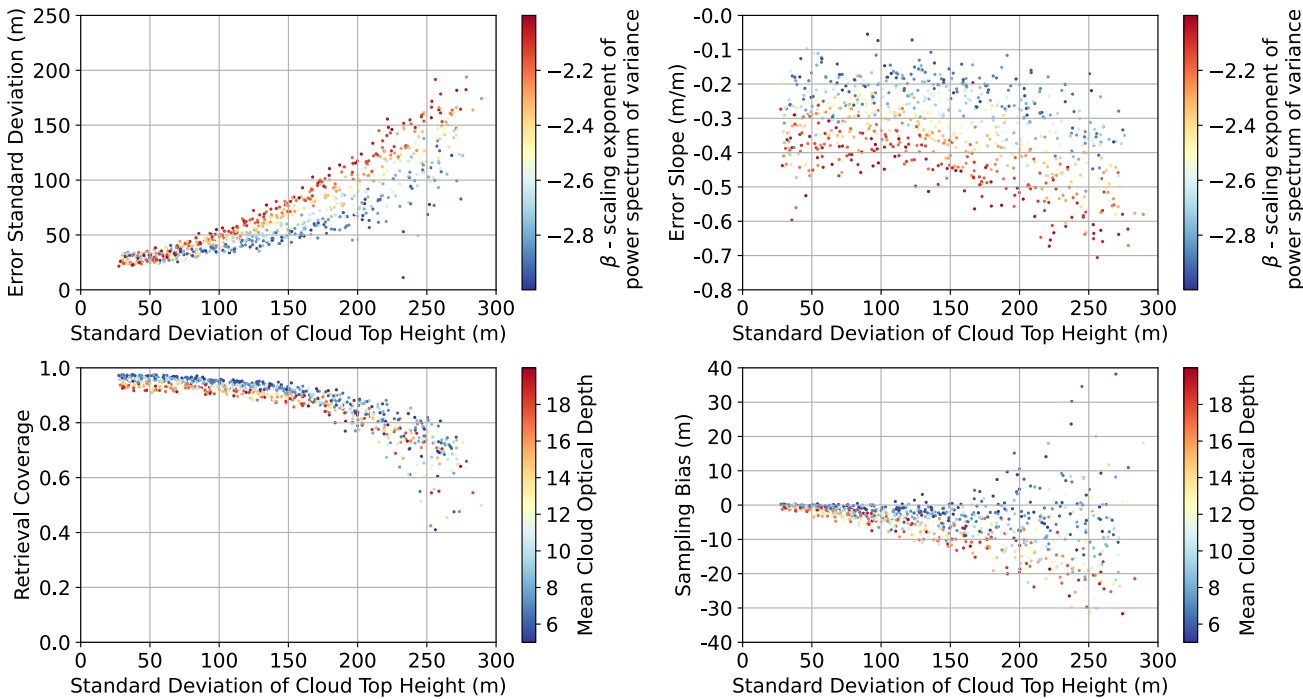

**Figure 8. Top Left: The variation of the stereo retrieval error standard deviation with the standard deviation of cloud top height in
each cloud field. Each dot represents a cloud field and is coloured by the slope of the power law describing the scale-distribution of
spatial variability. Top Right: As above but for the Error Slope. Bottom Left: As above but for the sampling bias, where each
cloud field is coloured by the mean cloud optical depth. Bottom Right: As above but for the retrieval coverage; the fraction of
cloudy pixels with successful stereo height retrievals. All results are for 50 m resolution observations using the 26.1° and 0°
backscattering viewing angle pair at a wavelength of 0.86 $\mu m$.**

The stereo-opacity bias varies weakly with the standard deviation of the cloud top height (Fig. 9) and $\beta$ (not shown).
Interestingly, mean cloud optical depth has a relationship with the stereo-opacity bias that has greater explanatory power at





more oblique solar zenith angles (Fig. 9). This feature is even more apparent at wavelengths with stronger absorption (not

shown). At a solar zenith angle of 30° there is essentially no relationship between the two variables for conservative scattering apart from a tendency for the most-negative biases to be associated with the thin clouds. At larger solar zenith angles it becomes clearer that the cloud optical depth appears to explain the variability in the stereo bias within two independent branches. This highlights the fact that optical thickness alone is insufficient to explain the stereo opacity bias, and that other characteristics of the cloud are involved.


To explain the cloud-dependence of the stereo-opacity bias, we introduce two different independent characteristics of the cloud-edge extinction field. The first is the vertical geometric distance from cloud top to an optical path of unity, $\Delta z_{\tau=1}$, which is closely related to Volkmer et al. (2024)'s definition of cloud top height. This quantity is highly anti-correlated with the cloud-edge extinction coefficient as geometric and optical distance are proportionally related by the volume extinction

coefficient, and the volume extinction coefficients of cloud are large enough that an optical path of unity is reached in the vicinity (20 m to 200m) of cloud edge (Kokhanovsky, 2004). The second characteristic of the cloud-edge extinction field is the relative vertical gradient of the extinction field at cloud edge. This quantity is computed as

$$\sigma_{grad} = \frac{1}{\sigma}\frac{\partial \sigma}{\partial z}\bigg|_{\text{cloud edge}} = \frac{\sigma_i - \sigma_{i-1}}{\sigma_i(z_i - z_{i-1})}$$

, where $i$ denotes the highest grid point with non-zero extinction (cloud top), and $i-1$ denotes the grid point below that.

Due to its normalization, this feature describes the shape of the extinction profile, not the magnitude of the gradient. We choose these metrics as we hypothesize that the stereo-opacity bias will be smaller when $\Delta z_{\tau=1}$ is small, as variability that is optically deeper into the cloud will be attenuated. We also hypothesize that the stereo opacity bias will be larger when $\sigma_{grad}$ is large, as a relatively low cloud-edge extinction with an increase towards the interior will increase the likelihood of a feature in the interior of the cloud producing the dominant spatial texture in the radiance imagery.


We compute each of these metrics from the cloud-field average of the variation of extinction from cloud-top. In other words, there is one value of $\Delta z_{\tau=1}$ per cloud field. We use the cloud-field average of these two features as we are searching for the simplest explanation for the variation of stereo opacity bias with cloud-type. The volume extinction coefficient is a spectral quantity, so these measures vary by wavelength.





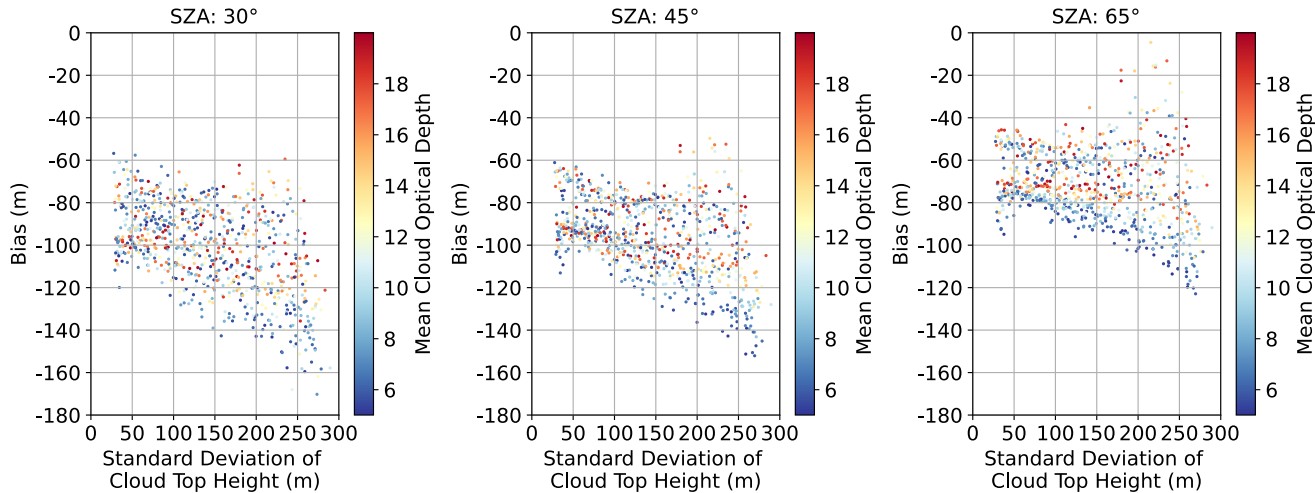


**Figure 9. The variation of the bias in the stereo-retrieved cloud top heights with the standard deviation of cloud top height. Each dot represents a cloud field and is coloured by the mean cloud optical depth. The columns show the different solar zenith angles. All results are for 50 m resolution observations using the 26.1° and 0° backscattering viewing angle pair at 0.86 $\mu m$.**

We quantify the effectiveness of these two predictors along with the mean cloud optical depth at predicting the stereo opacity bias by testing their skill using non-linear regression, specifically a random forest regression. Table 1 shows the r-squared value of each random forest regression at explaining the stereo opacity bias. The regression is applied independently at 50 m resolution across view angles and the r-squared value is averaged across all viewing angles. The random forest that utilizes both features explains a much larger portion of the variance (up to 70%) than the models that use only one (~40%).

The two predictors perform worse individually, but at a comparative level, demonstrating that they provide independent information about the origin of the stereo opacity bias. The mean cloud optical depth still adds skill in addition to these two features of the cloud-edge extinction field, increasing r-squared values up to maximum of 81%, especially for smaller solar zenith angles. This is expected because the total cloud optical depth changes the intensity of the background radiance field, which modifies the contrast with which cloud-top features can be observed (Davis et al., 2021a).




**Table 1. The r-squared value of random forest regressions at predicting the bias in the stereo retrievals using different combinations of $\sigma_{grad}$ , $\Delta z_{\tau=1}$ and $\bar{\tau}$ as predictors. Results are shown for all different combinations of wavelength and solar geometry. Results are similar across all viewing zenith angles.**

| Solar Zenith Angle | Predictors | Wavelength | | | | |
|---|---|---|---|---|---|---|
| | | 0.86 $\mu m$ | 1.65 $\mu m$ | 2.11 $\mu m$ | 3.75 $\mu m$ | 11 $\mu m$ |
| 30° | $\sigma_{grad}, \Delta z_{\tau=1}$ | 0.72 | 0.7 | 0.68 | 0.55 | 0.62 |
| | $\Delta z_{\tau=1}$ | 0.43 | 0.43 | 0.41 | 0.39 | 0.42 |
| | $\sigma_{grad}$ | 0.55 | 0.54 | 0.53 | 0.38 | 0.43 |
| | $\sigma_{grad}, \Delta z_{\tau=1}, \bar{\tau}$ | **0.81** | **0.81** | **0.78** | **0.65** | **0.73** |
| 45° | $\sigma_{grad}, \Delta z_{\tau=1}$ | 0.78 | 0.76 | 0.74 | 0.65 | 0.65 |
| | $\Delta z_{\tau=1}$ | 0.48 | 0.5 | 0.5 | 0.46 | 0.43 |
| | $\sigma_{grad}$ | 0.63 | 0.61 | 0.59 | 0.43 | 0.46 |
| | $\sigma_{grad}, \Delta z_{\tau=1}, \bar{\tau}$ | **0.87** | **0.88** | **0.87** | **0.85** | **0.8** |
| 65° | $\sigma_{grad}, \Delta z_{\tau=1}$ | 0.7 | 0.69 | 0.65 | 0.55 | 0.67 |
| | $\Delta z_{\tau=1}$ | 0.43 | 0.45 | 0.41 | 0.3 | 0.44 |
| | $\sigma_{grad}$ | 0.57 | 0.54 | 0.5 | 0.41 | 0.47 |
| | $\sigma_{grad}, \Delta z_{\tau=1}, \bar{\tau}$ | **0.85** | **0.86** | **0.85** | **0.83** | **0.8** |


## 3.4 Accuracy of paired retrievals

The large relative contribution of systematic bias to the overall error budget in the stereo retrievals (Fig. 3 & 7) suggests that two, time-differenced sets of stereo retrievals may be highly precise when detecting a change in cloud top height over a short time interval. A Tandem Stereo Camera was proposed as part of NASA's Aerosol Cloud Convection Precipitation (ACCP)

study (Braun et al., 2022) to exploit this concept, and it was included in the original designs for the NASA Atmospheric Observing System. The design included two satellite platforms that carried a set of tandem stereo imagers. The tandem-pair design allows for CTH to be determined independent of cloud motion from near-simultaneous image pairs, and for AOS the image pairs would have been collected approximately 45 seconds apart with viewing angle pairs of (+38.1°, 0°) and (-38.1°, 0°). Using simulated satellite visible imagery based on high resolution (20 m) large eddy simulations, it was found that 40 m

satellite imagery at these viewing geometries could be used to obtain CTH in each image pair with roughly 30 m (RMS) uncertainty and a vertical velocity better than 1 m/s (RMS) (personal communication, Roger Marchand).





Inspired by the Tandem Stereo Camera concept, we study the differences in the cloud top height retrieved by forward and backward scattering viewing pairs for our wide ensemble of static cloud fields. Accurate retrievals of vertical velocity rely on the cancellation of systematic errors between cloud-top height retrievals by two different viewing pairs. We provide a broad perspective on this phenomenon by analysing a wide variety of clouds and solar geometries. We emphasize that our analysis only provides a lower-bound on the accuracy of retrieving time-differences in cloud-top height. This is because temporal variability in cloud properties can cause changes in radiance features and there is also the need to simultaneously retrieve horizontal displacement of features in the time-varying case. Both factors will further reduce retrieval accuracy.

We note that one of the science goals for retrieving time-differenced cloud top height is the inference of cloud-top vertical velocities for clouds that are not detectable by radar. However, great care must be taken in interpreting the temporal variation of cloud top height as an atmospheric velocity as sources and sinks of cloud water will decorrelate the relationship between the two (Dandini et al., 2022a). We do not consider such effects as our cloud fields are strictly static. So, again, our estimates are lower bounds on the accuracy of stereoscopic methods for retrieving cloud dynamics.

The ACCP Science & Traceability matrix (https://aos.gsfc.nasa.gov/docs/ACCP_SATM_Rel_Candidate_G.pdf) stated a desired precision of the change in cloud top height of 1 ms$^{-1}$ when averaged over a $(500 \text{ m})^2$ region. We interpret this uncertainty at the 1-sigma level, i.e., 68% of errors should be smaller than this. For the Tandem Stereo Camera design, there is a time separation of 45 seconds between stereo retrievals, requiring a precision of 45 m between retrievals over the $(500 \text{ m})^2$ regions to meet the desired uncertainty of 1 ms$^{-1}$. Instrumental uncertainties must also be accommodated within this budget along with the uncertainty in the matching. Thankfully, unlike radiometric techniques for cloud top height retrieval, there are no radiometric calibration uncertainties or sensitivity to ancillary data about atmospheric and surface temperatures, trace gas concentrations, surface emissivity, and parameterizations of gas optics. This means that the dominant source of uncertainty in the retrieval is the geometric registration of the cameras. If we assume an additional 0.25-pixel (12.5 m) uncertainty in the matching due to inter-camera geometric registration uncertainty, the precision with which a cloud top height retrieval should be performed should be 30 m, assuming a 50 m pixel resolution. We also examined the relevance of instrumental noise and found it to have a negligible effect on the matching process due to the high signal-to-noise ratios of visible imagery of the clouds, which excludes sub-visible cirrus. The actual proposed spatial resolution for the Tandem Stereo Cameras was 40 m so the design will be a little more precise than our simulations.

We find that the median absolute difference between the stereo retrievals is reliably less than the 30 m target precision when the standard deviation of the cloud top height is less than 200 m and when sun glint is not observed (Fig. 10). The quantification of the error with the median demonstrates that at least 50% of the retrievals are this precise. Root-Mean-Square errors are slightly larger, indicating the presence of some positive skew in the error distribution. The issues due to sun glint occur for the solar zenith angle of 30° when the relative azimuth angle is close to 180° so that the forward scattering



viewing angle is very close to the specular reflection from the ocean surface. The resulting errors are largest for the optically thinnest clouds (not shown). Performance is similar at other wavelengths, except the sunglint issue is several times larger at 3.75 $\mu m$ and absent at 11 $\mu m$ (not shown).


**Figure 10. The variation of the median absolute difference between stereo retrievals using (+38.1°, 0°) and (-38.1°, 0°) height where they both have valid retrievals with the standard deviation of the cloud top height. Each dot represents a cloud field, and they are coloured by the relative azimuthal angle between the solar and viewing planes. The different rows have different solar zenith angles (SZA). Bear in mind the change in colorbar scale with solar zenith angle. The left column shows retrievals at native 545 resolution. The right column shows the median absolute differences once the differences are averaged to 500 m resolution. All retrievals use the 0.86 $\mu m$ wavelength.**

The differences between the stereo height retrievals are not entirely random. They include a systematic component due to incomplete cancelling of the stereo-opacity biases when comparing forward and backward scattering observations (Fig. 11). 550 When interpreted as a rate of change in cloud top height, this will appear as a "drift" of the clouds up or down. This




component tends to be small but may dominate regional or seasonal uncertainties if the residual systematic error varies systematically with solar geometry or cloud type. Figure 11 shows that the cloud-field-mean of the cloud top height differences may change by as much as 30 m with solar zenith angle for the paired observations, which is slightly larger than the equivalents for the unpaired observations (cf. Fig 3). Outside of the sun-glint regions, the drift can vary by as much as 20

m with mean cloud optical depth. The results suggests that the regional average of the rate-of-change of cloud top height may only be determined with a precision of ~0.67 ms$^{-1}$ using a Tandem Stereo Camera design (i.e., 30 m / 45 s). This systematic error feature would be most relevant for long-term change detection.

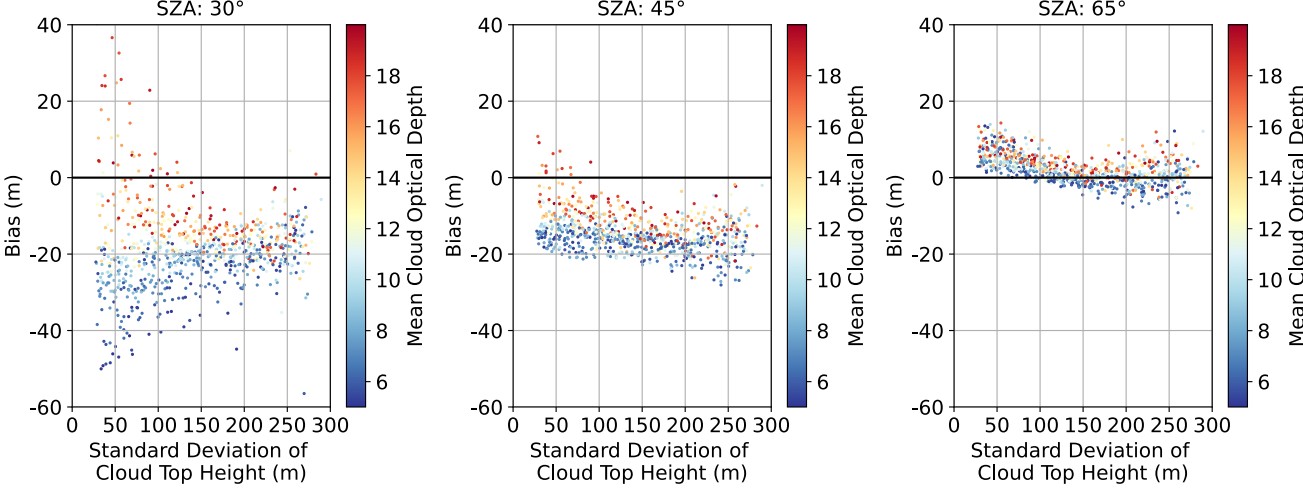

**Figure 11. The variation of the mean difference between stereo retrievals using (+38.1°, 0°) and (-38.1°, 0°) where they both have**
**valid retrievals with the standard deviation of the cloud top height. Each dot represents a cloud field, and they are coloured by the mean cloud optical depth. The different panels have different solar zenith angles (SZA). All retrievals use the 0.86 $\mu m$ wavelength.**

The retrieval coverage is also lower for the paired retrievals, as expected, ranging from ~95% for less bumpy cloud fields to ~40% for very bumpy cloud fields, with lower shared coverage of ~25% in sunglint conditions (Fig. 12). This means that a large percentage of the population of successful retrievals will meet the 30 m precision requirement at 500 m, even if

different cloud top height standard deviations are sampled with uniform probability. However, the resulting sampling biases may be particularly significant, depending on the scientific application. One of the main motivations to measure cloud vertical velocities is to constrain convective mass fluxes (Tan et al., 2018), which are contributed preferentially by larger clouds due to their larger updraft area. If larger clouds are also those that are bumpiest at an 8 km scale, then our results indicate that there will be a strong sampling bias in any estimate of mean mass flux, for example. This highlights the

importance of assessing the representativeness of the simulation results presented here to develop a more complete understanding of the error characteristics of actual stereoscopic retrievals.




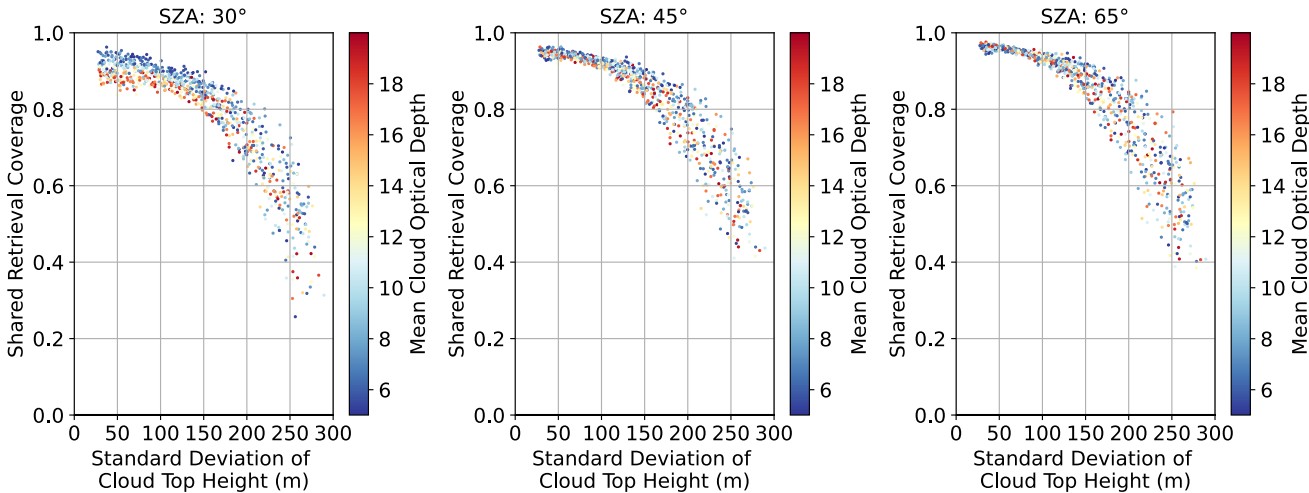

**Figure 12. The variation of the fraction of cloud with valid stereo retrievals from both the (+38.1°, 0°) and (-38.1°, 0°) camera pairs with the standard deviation of the cloud top height. Each dot represents a cloud field, and they are coloured by the mean cloud**
**optical depth. The different panels have different solar zenith angles (SZA). All retrievals use the 0.86 $\mu m$ wavelength.**

### 3.5 How bumpy are clouds?

Due to the strong variation of stereo retrieval accuracy and coverage with the standard deviation of cloud top height, it can be uncertain exactly how well the method will perform on real clouds without knowledge of the probability distribution of the standard deviation of cloud top height for different cloud types. Stratocumulus clouds are known to be flat, especially at
small scales, with typical standard deviations of cloud top height over roughly a 8 km interval of between 30 m and 70 m (Boers et al., 1988; Loeb et al., 1998). Clouds tend to become bumpier when they are more cumuliform and can reach standard deviations larger than 200 m over 8 km intervals for boundary layer clouds (Loveridge & Di Girolamo, 2024). To supplement these data, which didn't focus on cumulus-only regions, we analyse measurements from the CAMP[2]Ex field campaign near the Philippines (Reid et al., 2023). The airborne lidar data show that 75% of the shallow marine cumulus had
standard deviations of cloud top height over roughly 8 km intervals that were less than 200 m (Fig. 13).



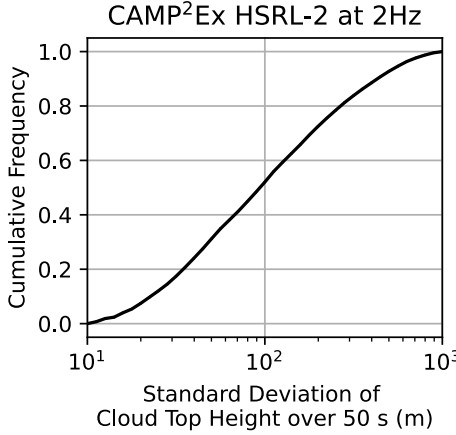

**Figure 13. The cumulative frequency of the standard deviation of cloud top height computed along 50 s flight segments (~8 km). Data are from the HSRL's 2 Hz cloud top height retrieval, as processed in Fu et al. (2022), while it was deployed on the P-3 aircraft for the CAMP²Ex field campaign in the Philippines. Flight segments are filtered to only include those with maximum cloud top heights below 3 km, resulting in sampling primarily shallow cumulus clouds.**

We can use this statistic to extrapolate our simulation results to the cloud types sampled during CAMP²Ex. We assume that the relationship between the standard deviation of cloud top height and the stereo retrieval performance generalizes to these clouds. The critical component of this assumption appears to be that the assumed scale-dependence of the cloud-top height variability is realistic, given the sensitivity of the stereo retrieval accuracy to cloud-top height variability at a similar scale to the instrument resolution. Unfortunately, we cannot unambiguously study CTH variability at such scales, due to the horizontal resolution of the lidar data (65 m to 90 m).

Based on this extrapolation, 75% of shallow cumulus clouds are expected to occur within the range for high-precision retrievals (< 30 m paired accuracy) with good shared retrieval coverage (> 65%). Trivially, almost all stratocumulus clouds are expected to occur in the range for high-precision retrievals with shared coverage greater than 90% outside of sunglint conditions. We didn't find a significant relationship between the standard deviation of the cloud top height over the 50 second window and the median cloud transect length in the CAMP²Ex observations (not shown), suggesting that any dependence of cloud-top height retrieval accuracy on cloud-top bumpiness will equally affect cumulus clouds of differing sizes, at least for shallow cumulus clouds that are well resolved by the horizontal resolution of the lidar measurements (65 m to 90 m).

## 4 Discussion

Our simulations have quantified several important error characteristics of stereoscopic methods for retrieving cloud top height. First, we have identified that stereo retrievals tend to be negatively biased on the order of -100 m. The stereo-opacity



bias has a strong dependence on the structural features of the cloud field, particularly the magnitude and shape of the extinction profile near cloud edge. When the cloud-edge extinction starts small and increases rapidly with depth into the cloud, the stereo-opacity bias is its most negative, reaching as negative as -175 m. The stereo-opacity bias also tends to decrease in magnitude, becoming less negative, as absorption increases and viewing zenith angle increases. Second, we have identified that stereo cloud top height retrievals are smoother than the truth, an error that also decreases in magnitude as absorption increases, viewing zenith angle increases, and spatial resolution increases. This error feature also depends on cloud structure, being more significant when the proportion of cloud-top height variability at smaller scales is larger. Third, we identified that the retrieval coverage is lowest for the wavelengths with the most anisotropic escaping radiation (3.75 $\mu m$) and is highest for the wavelength with the weakest anisotropy (11 $\mu m$).

To begin our explanation of these behaviours we focus on the 11 $\mu m$ results, which are unique in that they are emissive and have weak scattering. In this scenario, the physics of the radiative transfer are simplified, providing a useful starting point for our discussion. We note that the 11 $\mu m$ stereo retrievals rely on the existence of temperature variability to create textures in optically thick clouds. This is true for our clouds, which have a roughly linear variation of temperature through the cloud layer. There may not always be sufficient temperature variability for the 11 $\mu m$ stereo retrievals to retrievals, such as when clouds form near an inversion layer. With that caveat in mind, when the volume extinction coefficient is small near cloud edge but increases with depth into the cloud, the emission by the larger extinction variability further into the cloud can still escape without being substantially attenuated, resulting in features in the radiance imagery. The deeper into a cloud the feature, the larger it must be to escape despite the exponential nature of attenuation. The limits on the magnitude of spatial variability in cloud microphysical properties allowable by cloud physics and turbulent dynamics therefore provide the upper limit on the stereo-opacity bias. The intermittence of turbulence allows substantial spatial features in the cloud microphysical properties, and hence optical properties to exist (Davis et al., 1999; Marshak et al., 1997). However, these features would have to be orders of magnitude larger than their surroundings to be detectable in the radiance field at large optical paths (e.g., 10) from cloud edge (Davis et al., 2021b). This simple fact is what limits the stereo-opacity bias to be relatively small in optically thick clouds. However, in the quasi-linear regime of small optical thicknesses, features from large geometric distances from cloud edge can produce the dominant spatial texture.

The tendency for the retrieved cloud top height field to be smoother than truth, as quantified by the error slope (Eq. 1), is very small at 11 $\mu m$, on average, but increases as cloud-top-height variability is found at smaller spatial scales. We can attribute this to the impact of the regularization in the MGM algorithm compensating for the presence of significant variability in cloud-top-height within the matching region. As an image correspondence becomes ambiguous, the regularization will bias the retrieval towards a smooth solution. This mechanism also explains the increase in smoothing error when more oblique views are matched with nadir, as the assumption of a single disparity in the matching window becomes less accurate.



As we move to the solar spectrum, we introduce anisotropic scattering to all these mechanisms. Scattering helps to

communicate features from deeper into cloud to the radiance imagery, albeit in a smoothed form (Davis et al., 1997, 2021a; Kokhanovsky, 2004; Marshak et al., 1995; Platnick, 2001). Interestingly, the forward scattering peak of the Mie phase function counteracts this smoothing by focusing scattered light in a peak that helps preserve spatial contrast (Kokhanovsky, 2004; Loveridge et al., 2023a). The increased likelihood of a deeper feature being visible in the imagery as a result of scattering results in an increasingly negative stereo-opacity bias as single scattering albedo increases. The closer the single-

scattering albedo is to unity, the greater the contribution of these deeper, smoothed features will be to the radiance imagery. The presence of many features corresponding to different depths into the cloud lowers the signal-to-noise ratio in the matching process. As a result, the stereo matcher will tend to retrieve stereo heights that are smoother due to regularization. Increasing the single-scattering albedo at solar wavelengths also decreases the anisotropy of the escaping radiation, as higher-order scattering events are more common when absorption is weak. Importantly, the more anisotropic the exiting

radiation is, the less likely it is that the spatial textures will correlate across views. The spatial textures that correlate best across angles will be those that are less anisotropic, originate from higher-order scatter, and therefore form deeper into the cloud. This mechanism explains the feature that retrieval coverage is worst at 3.75 $\mu m$ and that the smoothing error tends to increase at shorter wavelengths and larger viewing zenith angles. These insights into the physical controls on the accuracy of stereo height retrievals lead to several important implications in the use of stereoscopic retrievals of cloud top height

retrievals.

We can expect the stereo-opacity bias to vary with cloud morphology, thickness and therefore cloud life cycle. For example, a small cloud-edge extinction coefficient with an increase in the cloud interior may be characteristic of a dissipating cloud, differing from active convective thermals. The resulting change in the stereo-opacity bias between cloud types may be

misconstrued as signal. There are various contexts in which this systematic error is important. The strong contribution of systematic errors to the overall error budget of the stereo retrieval is an important factor to consider when assessing cloud-top-height trends from MISR or other similar instruments as spurious trends in cloud top height can arise due to changes in cloud morphology and optical thickness. Similarly, appropriate consideration of systematic error in the stereoscopic retrieval should be given when measuring differences between cloud types.


The stereo-opacity bias should be considered when deriving 3D cloud geometry to improve the interpretation of radiances in terms of cloud microphysics. Tomographic retrievals of 3D cloud microphysics using 3D radiative transfer can benefit from information about 3D cloud geometry to initialize or constrain the retrieval (Doicu et al., 2022; Levis et al., 2020; Loveridge et al., 2023b). The quantification of the stereo opacity bias presented here indicates that a buffer region of ~100 m should be

added around the volume indicated by the stereoscopic retrievals when producing a volumetric cloud mask for a



tomographic retrieval. Alternatively, a volumetric cloud mask from stereo should be considered as an estimate of the confidently cloudy volume, but not the complete extent of a cloud volume.

The stereo retrievals at 50 m resolution meet the desired accuracies for retrieving cloud top height for assessing cloud radiative feedbacks (Ohring et al., 2005), mostly regardless of wavelength choice. The wavelength-choice of the stereo retrieval errors is more likely to be governed by the sensor cost, isolation of cloud from surface signals over land or in sun-glint, retrieval behaviour in optically thin cases ($\tau < 5$), and signal-to-noise ratio. These factors were not investigated in this study and will certainly influence the choice of wavelength (along with cost) and are ripe areas for further investigation. Our results do show that 11 $\mu m$ radiances can provide stereo retrievals with slightly smaller systematic errors than visible

retrievals for optically thick clouds over ocean, providing another option for stereoscopic retrievals. High-resolution stereo retrievals may provide a useful constraint on the sounding of atmospheric temperature and water vapor in the planetary boundary layer (Martins et al., 2010). This is because clouds act as tracers for sharp transitions in the thermodynamic structure of the boundary layer.

Our results support the retrieval of changes in cloud top height from paired stereo retrievals. This is because of the cancellation of systematic errors between two views. Further work is needed to confirm that the cloud-edge extinction field possesses a sufficiently similar structure during short time intervals (~45 s) so that the stereo-opacity bias cancels between two time-differenced views. Work with LES simulations discussed above indicates that this assumption does hold (personal communication, Roger Marchand).


When we compare our results to those of other model-based studies such as Dandini et al. (2022) and Volkmer et al., (2024), we find differences in the quantification of the stereo-opacity bias that are worth exploring and explaining. These two studies used cloud fields derived from several different Large Eddy Simulations (LES). Our study showed a significant sensitivity of stereo matching accuracy to the representation of the cloud structure at small scales (~150 m). Due to numerical diffusion,

the effective resolution of LES tends to be a factor of 4 to 5 coarser than the grid spacing, so that a grid spacing that is 4 to 5 times higher than the instrument resolution is required to ensure that the cloud-top height field is not overly smooth at the scale of the matching window (3 pixels or ~150 m). In Dandini et al. (2022), the imager resolution was higher than, or comparable to, the grid spacing of the fluid dynamical simulations, resulting in small variations in ground-truth cloud top height within the matching window. This may explain why the matching precision was a factor of 2 to 3 better than in

Volkmer et al. (2024) and in this study for more cumuliform clouds. Alternatively, it could also be that Dandini et al.'s evaluation metric (M3C2), evaluates precision over a coarser region resulting in the cancellation of some errors. The study of Volkmer et al. (2024) utilized very high-resolution LES simulations (10 m by 10 m by 5m) designed to maximize the fidelity of small-scale features. Their results have a similar precision to the range explored in our simulations and have a bias of -46 m. However, they define their ground truth with respect to an optical path of unity within the cloud rather than cloud edge.



On average this corresponds to a further 20 m to 30 m into the cloud for a bias of -66 m to -77 m. Similarly, they also identified a smoothing error, where lower cloud edges had overestimated cloud top heights while high cloud tops were underestimated (Volkmer et al., 2024b). The corroboration of our results against cloud fields from LES supports our key conclusions.

The availability of a wide range of matching algorithms (Beekmans et al., 2016; Castro et al., 2020; Dandini et al., 2022b; Fisher et al., 2016; Foley et al., 2024; Kölling et al., 2019; Muller et al., 2002), and the need for algorithms tailored for application to cloud, suggest that a stereo matching algorithm development and intercomparison activity would be beneficial. We identified that ~25% of cumulus clouds sampled in CAMP$^2$Ex have large standard deviations of cloud-top height over 8 km (>200 m) for which stereo matching degrades significantly, with error standard deviations of ~200 m and retrieval

coverage as low as 40% despite the use of 50 m resolution measurements. It is therefore very important to continue to develop improved stereo matching retrievals to improve performance for these cloud types. The focus of an intercomparison activity would be in the development of regularization techniques that are informed by prior knowledge of cloud structures and their relationship to image features. In particular, the correlation between reflected intensity and height changes. The development and validation multi-spectral matchers may also be of benefit. Water clouds are also not the only volumetric

emitters or scatterers to which image matching algorithms may be applied. Matching algorithms are also applied for sensing of atmospheric motion using gas emission features (Lean et al., 2015; Mueller et al., 2017) or for studying aerosol plumes (Kahn et al., 2007). Such an intercomparison activity would also be of benefit for these applications.

## 5 Conclusions

We evaluated stereoscopic retrievals of cloud top height using synthetic imagery generated by applying a 3D radiative

transfer model to an ensemble of ~800 stochastically generated 3D cloud fields. The cloud fields varied in their fractional coverage, bumpiness, spatial organization, microphysics and optical depth. We examined the sensitivity of the stereo retrieval algorithm to wavelength and solar-viewing geometry as well as the structural characteristics of the cloud fields. We found that stereoscopic retrievals have an average bias of about -70 m to -100 m in the solar spectrum that can vary by cloud from -20 m and -175 m as the spatial gradient in the extinction field becomes steeper at cloud edge and clouds become

optically thinner. The bias is caused by the emergence of textures in imagery from extinction features within the cloud due to multiple scattering. By contrast, stereoscopic retrievals applied to 11 $\mu m$ imagery, where emission and absorption largely control the emerging radiation field, have small bias. The bias is similar across instrumental resolutions between 50 m and 250 m.

We found that stereoscopic retrievals tend to retrieve cloud top heights that are overly smooth. We quantified this effect using the slope of the regression between cloud top height error against the ground truth cloud top height in each cloud field,



which we term the error slope. The error slope decreases to -0.7 when there is strong cloud top height variability at spatial scales comparable with the matching window (3 pixels) for high-resolution (50 m) measurements. The magnitude of the smoothing decreases with stronger absorption and increases with viewing zenith angle. We ascribe this behaviour to two factors. The first is that the light that reaches the sensor tends to have undergone fewer scattering events on average, when the single-scatter albedo is smaller. The multiple scattering reduces the decorrelation between image features and cloud top height features. The second factor is algorithmic and involves the smoothing effect of regularization in the stereo matching algorithm when there is ambiguity in the stereo match due to variations in the cloud top height within the matching window. The smoothing error is the dominant control on the precision of the stereo retrieval. The standard deviation of the retrieval error varies from 25 m to 250 m as the standard deviation of cloud top height increases to 250 m for 50 m resolution observations. The systematic nature of the smoothing error produces a slower convergence of precision with increasing resolution than might be expected due to extrapolation of a pixel-relative matching accuracy.

We analysed the consistency of paired retrievals and showed that, due to the dominant contribution of the systematic bias to the error budget, paired stereo retrievals tend to agree to within 30 m more than 50% of the time when the standard deviation of cloud top height over a $(8 \text{ km})^2$ region is less than 200 m. This result supports the application of multi-platform stereo imagers for the retrieval of time-differenced cloud top heights to constrain cloud dynamics, as they imply precisions better than 1 ms$^{-1}$ for most clouds. Based on airborne lidar measurements of bumpy shallow cumulus, 75% of cloudy pixels fall in this category of cloud-top bumpiness, suggesting good precision and that the coverage of successful retrievals will be better than 80%. Our results show that there are rapid reductions in the number of successful retrievals and retrieval precision in cloud fields when the standard deviation of cloud top height over a $(8 \text{ km})^2$ region increases beyond 200 m.

Further improvements in matching algorithms with regularization schemes tailored to clouds will be beneficial to improve retrieval coverage and precision in these cloud types to ensure that stereo retrievals perform equally well in all regimes. Given the strong performance of stereoscopic cloud top height retrievals, we argue for an algorithm development and intercomparison activity targeted towards selecting the most effective means of retrieving cloud macrophysics and dynamics from multi-angle stereoscopy for deployment in space. We also highlight that the high precision of stereoscopic retrievals can provide an important constraint for sounding of the atmosphere using infrared or microwave measurements, especially in multilayered cloud regimes (Mitra et al., 2023). The stereo-opacity bias should be considered when combining stereoscopic retrievals with other remote sensing instruments or as a constraint on cloud volumes, especially when they are small. Highly accurate measurements of cloud boundaries from stereoscopic retrievals can be used as part of remote sensing retrievals of cloud microphysics that account for cloud geometry and 3D radiative transfer.



**Code availability**

The software to perform the analysis are archived on Zenodo along with data to reproduce the figures (Loveridge, 2024).

**Author contribution**

JL designed and performed the analysis under the advisement of LD and prepared the initial draft of the manuscript. JL and LD contributed to the editing of the manuscript.

**Competing interests**

The authors declare that they have no conflict of interest.

**Acknowledgements**

We would like to thank Roj Marchand for fruitful discussions on the Tandem Stereo Camera concept, Arka Mitra for discussions about stereo matching, and Frank Evans for making his SHDOM 3D radiative transfer code publicly available. LD would also like to acknowledge a former graduate student in his group, Daeven Jackson, for the many discussions they had on stereoscopic CTH errors, before his untimely death. His gentle heart and wonderful character is missed.

**Financial support**

JL gratefully acknowledges support from NASA's FINESST program (Grant 80NSSC20K1633). LD gratefully acknowledges support by the MISR project through the Jet Propulsion Laboratory of the California Institute of Technology (contract no. 1474871) and by NASA's ACCDAM program (Grant 80NSSC21K1449).

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
