# Peer review of "Errors in stereoscopic retrievals of cloud top height for single-layer clouds."

_EGUsphere, 2025_

## Author Response (AR1)

We thank the reviewers for their thoughtful and constructive comments. Our responses are embedded below in red. Quoted line numbers refer to the tracked-changes version of the manuscript.

Reviewer 1:

General comments:

The manuscript thoroughly assessed the stereo-opacity bias in an idealized case, i.e., the observations were represented by model equivalents simulated by SHDOM, a widely used radiative transfer solver, based on 3D cloud fields derived from a stochastic generator (Loveridge & Di Girolamo, 2024). The findings are robust since the stereo-opacity bias was explored under different circumstances covering different cloud coverage, cloud-top bumpiness, etc. The authors presented a solid study which helps the understanding of the stereoscopic retrievals by satellite imagery from visible to infrared wavelength. Therefore, I suggest the manuscript be published after some specific comments were properly addressed.

Specific comment 1:

Introduction, line 48-49. "This is due to their ability to achieve high (< 50 m) resolution and precision retrievals of cloud boundaries from multiple viewing angles". I checked the two references and found that the satellite data were gridded at a resolution of 2.9 m (or 58.1 m) in Castro et al. (2020) and of 50 ~ 200 m in Dandini et al. (2022b). My doubt is that the resolution of the retrieved CTH should be more related to the raw satellite data rather than stereoscopic retrieval technique. In my opinion, the spatial resolution of satellite instruments provides the high limit to the resolution of the retrieved products.

The reviewer is completely correct that the satellite imagery must be high-horizontal resolution to achieve high-precision in height retrievals from stereo. Our goal was to contrast this with other techniques such as infrared, that cannot achieve high vertical precision even if the imager resolution (horizontal resolution) is high due to their reliance on ancillary data and radiative transfer modelling. We have revised this sentence to make this point clear (Line 50-54).

Specific comment 2:

Methodology, Section 2.2. The periodic horizontal boundary conditions were used for the 3D radiative transfer simulations by SHDOM. It is a useful approximation to account for horizontal photon transports periodically near the boundaries of a finite domain, since an adequately large domain would cause heavy computational burden.

However, the radiative transfer simulations of incoming and outgoing photons in the horizontal directions should be less accurate near the domain boundaries. I was wondering whether the grids near the domain boundaries were discarded or not, could the authors please give an explanation of why?

Since we used periodic boundary conditions in the radiative transfer, our simulated clouds are therefore also horizontally periodic. In this sense, there is no additional errors incurred from this assumption near cloud boundaries (i.e., the stereo retrievals are as good at the domain boundaries as they are within the domain).   So we did not discard any of the data near the domain boundaries. We use all of the stereo retrievals, including those derived from pixels at the edge of each image. We have made note of this at the beginning of Section 2.2 (Line 180-181).

Specific comment 3:

Methodology, Section 2.3. The stereo matcher is a crucial step of the stereoscopic retrieval technique. The authors already introduced this part very well. However for a potential reader (like me) who is unfamiliar to this technique, this part is hard to understand. Therefore, it would be really helpful if the authors could provide an example showing how the stereo matcher works.

We have added a step-by-step description of the matching process to compute image disparities at the beginning of Section 2.3 (Line 213-219).

Specific comment 4:

Results, Section 3.4, line 491-494. I am confused how the results in Fig. 3 & 7 could led to the conclusion that two, time-differenced sets of stereo retrievals may be highly precise when detecting a change in cloud top height over a short time interval. An explanation would be really helpful and appreciated.

Our apologies, this should be Fig. 3 & **6**. When errors are systematic (Fig. 3), they can be common to the two sets of stereo retrievals. Then, when the difference between the two retrievals is taken, they cancel. Then, the difference in cloud top height is known more precisely than the absolute value of the cloud top height. Random errors, on the other hand (Fig. 6) will tend to add in quadrature, so that differences in cloud top height are less certain than either cloud top height. Figure 3 shows that systematic errors are large compared to random errors shown in Figure 6, so that we might expect differences in cloud top height to be retrieved more precisely.

We have added clarification of this line of reasoning at the lines quoted by the reviewer (Lines 529-532).

Specific comment 5:

The results of this study are based on idealized cases with a lot of approximations, including the sea surface with a pre-defined reflectance of the MODIS 0.86 um channel is 0.0531 or 0.0594, the limited solar zenith angles (30°, 45°, and 60°), and stratocumulus clouds with flat cloud bases. I have two major concerns to extend the findings to real-world cases.

To clarify, the reflectances of 0.0531 or 0.0594 are the reflectances used for cloud masking, not a preset reflectance of the sea surface. The sea surface is treated by a BRDF model as described at Line 192. But yes, this is constant for all simulations.

1) The robustness of the findings could be further strengthened by including more solar zenith angles since 3D radiative transfer simulations are highly dependent on the solar zenith angles. But I guess including more solar zenith angles could make the 3D radiative transfer simulations rather time-consuming. If so, I would suggest mention this limit in this part or other parts.

We expect the performance of the stereo retrieval to be smooth with solar zenith angle, based on the fact that spatial textures in the radiance imagery that are used by the stereo retrieval are smooth with solar zenith angle (Várnai, 2000). As a result, we expect the results to extrapolate and interpolate beyond the examined solar zenith angles. We have added this argument and discussion of the limitation of the set of solar zenith angles to the Discussion (Section 4, Line 709-711. We have not performed additional simulations due to the substantial computational expense.

2) The study only discussed liquid water clouds. I am also curious how the findings would change if the stereoscopic retrieval was applied to ice clouds. For the climate research or precipitation processes, the macro-physical properties such as CTH should be also important. I would like to see some discussions on this topic in the revised manuscript.

The reviewer is correct that ice clouds are important and that the relevance of our results to ice clouds should be discussed. We now include discussion on this topic in Section 4 (Lines 712-714 & 746-748. We expect our results to generalize to optically thick ice clouds. Due to the weaker anisotropy of rough ice particle phase functions, we expect similar or better performance when clouds are thick. However, optically thin ice

clouds such as cirrus are also very important, and our results will not generalize to them. We highlight this as an important area of future study (Lines 748).

Reviewer 2:

In my opinion, this is an excellent paper that will provide helpful guidance to the community in designing instruments and data processing algorithms for stereoscopic retrievals of cloud top height. The methodology is sound, and the presentation is of a high quality. A particular strength of the manuscript lies in the in-depth explanations and discussions it offers. Even so, the manuscript needs some minor improvements. Please find my specific comments below.

**Contents:**

Lines 134, 327-328, and Section 3.5: I wonder if it is important that the paper compares cloud top height variability statistics obtained differently for observed clouds than for simulated clouds. Specifically, for clouds observed during the CAMP[2]Ex campaign, the paper calculates the standard deviation of cloud heights encountered along an 8 km long straight line—whereas for simulated clouds, the paper uses the standard deviation of all cloud height values within 8 km by 8 km areas. It may be worth adding a brief note into Section 3.5 about whether the comparisons would look different if (instead of calculating an overall standard deviation of all cloud height values within a simulated field) we calculated cloud height standard deviation values for several individual 8 km long transects and then we used the mean of these standard deviation values to characterize cloud top height variations within each simulated cloud field.

We compared standard deviations of cloud top height estimated from 1D transects vs. the full area (8 km)$^2$ for the synthetic clouds. On average, the standard deviation of cloud top height from 1D transects is only ~15 m smaller. Based on this good agreement we can be confident in the comparison with the airborne data. We have added a note to this effect in Section 3.5 at Line 631-634.

Lines 133-149: It would help to clarify how the geometric and optical thicknesses of individual cloud columns are related to each other. Within an individual cloud field, do geometrically thicker clouds tend to have larger (or smaller) optical thicknesses and/or column-average extinction coefficients?

It is correct that geometric thickness is correlated with optical thickness. This is now stated explicitly (Line 154).

Lines 289-291: If my interpretation is correct, the wording should be changed to clarify that the sampling bias is the difference between the true heights of two sets of pixels (identified by colors other than gray or dark blue in Figs. 1b and 1d, respectively). This is needed because the current wording suggests that the sampling bias includes the

differences between the non-zero height values shown in Figs. 1b and 1d. (The difference between the values in Figs. 1b and 1d would not provide the sampling bias, as Fig. 1d shows retrieval results that are affected by retrieval errors as well as sampling issues.)

Thank you for catching this, the reviewer is correct. We have revised the corresponding sentence to clarify this (Line 322).

**Presentation:**

Line 193: It would help to explain what four relative azimuth values are the possibilities when selecting an azimuth value for each cloud and solar zenith angle.

We have added the list of the solar azimuth angles (Line 208-211). They were chosen so that the 3DRT simulations can also fulfill a different purpose. Specifically, so that the different oblique views sample scattering angles at the peak and edge of the rainbow.

Lines 209-211: This sentence contains the words "utilize" and "which" twice; one of the occurrences of each word should be replaced.

Done.

Line 248: I suggest replacing "Errors statistics" by "Error statistics".

Done.

Line 346: I recommend changing "As above" to something like "As in the left panel".

Done.

Lines 480-481: I suggest replacing "comparative" by "comparable". Also, it is unclear how providing a comparable amount of information implies that the information provided by the two predictors is independent.

We have made this change and clarified that it is the fact that the r-squared increases when used together that demonstrates that the two predictors provide independent information.

Figure 10: The caption should clarify what the grey and black lines (at 30 and 45 m, respectively) represent.

Done.

Line 582: The reference Loveridge and Di Girolamo (2024) is missing from the reference list.

Added, thank you for spotting this.

Lines 609-610: I recommend refining the wording, as extinction profiles do not have a magnitude.

Done.

Line 717: I recommend changing "cloud" to "clouds".

Done.